# VerbalTS: Generating Time Series from Texts

**Shuqi Gu** [1]   **Chuyue Li** [1]   **Baoyu Jing** [2]   **Kan Ren** [1]

## Abstract

Time series synthesis has become a foundational task in modern society, underpinning decision-making across various scenes. Recent approaches primarily generate time series from structured conditions, such as attribute-based metadata. However, these methods struggle to capture the full complexity of time series, as the predefined structures often fail to reflect intricate temporal dynamics or other nuanced characteristics. Moreover, constructing structured metadata requires expert knowledge, making large-scale data labeling costly and impractical. In this paper, we introduce VERBALTS, a novel framework for generating time series from unstructured textual descriptions, offering a more expressive and flexible solution to time series synthesis. To bridge the gap between unstructured text and time series data, VERBALTS employs a multi-focal alignment and generation framework, effectively modeling their complex relationships. Experiments on two synthetic and four real-world datasets demonstrate that VERBALTS outperforms existing methods in both generation quality and semantic alignment with textual conditions. The project page is at https://seqml.github.io/VerbalTS/.

## 1. Introduction

Time series modeling plays a crucial role in modern society, with applications spanning finance (Gao et al., 2024), medicine (He et al., 2023; Chen et al., 2024b; Jarrett et al., 2021), climate (Jing et al., 2021; 2022; 2024b;c), and energy (Lai et al., 2018). However, challenges like complex data collection and the rarity of extreme conditions limit access to large-scale, high-quality time series data. To address this, research has explored generating time series from

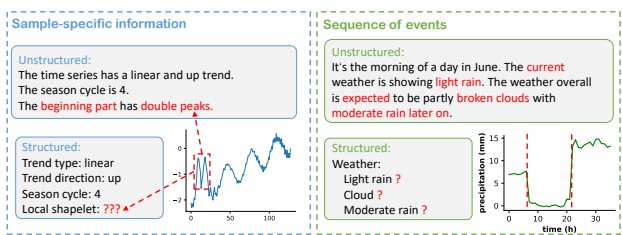

*Figure 1.* Illustration of the different control conditions. Compared to structured conditions, unstructured conditions are better at capturing unique, sample-specific information, e.g., special local shapelets that are not present in all samples (left figure). Additionally, structured conditions fail to effectively describe the sequence of events, a common characteristic of time series (right figure).

scratch (Narasimhan et al., 2024) or modifying existing ones (Jing et al., 2024a), enabling richer datasets and advancing time series modeling.

Current time series generation tasks can be broadly classified into unconditional (Desai et al., 2021; Yuan & Qiao, 2024; Pei et al., 2021; Yoon et al., 2019) and conditional generation (Wang et al., 2023; Lee et al., 2023; Coletta et al., 2024). Unconditional generation involves randomly sampling from the learned time series distribution, offering minimal control over the properties of the generated samples. In contrast, conditional time series generation usually leverages *structured* conditions such as metadata (Narasimhan et al., 2024), time series attributes (Jing et al., 2024a) or class labels (Li et al., 2022; Wang et al., 2023).

Generating time series from structured conditions is straightforward but comes with notable limitations. First, much of the real-world information about time series is *unstructured*, making the extraction of structured features from large, unorganized datasets both time-consuming and labor-intensive. Second, as illustrated in Fig. 1, this process often leads to significant information loss. For instance, sample-specific details that cannot be encapsulated within a uniform structure are frequently omitted. Similarly, sequential conditions that govern the chronological characteristics of time series may be inadequately captured by structured representations.

In this paper, we propose to generate time series using text descriptions as control conditions. Text, as a prevalent form of unstructured information, inherently conveys rich semantic content that allows for more nuanced and detailed ex-

---

[1]School of Information Science and Technology, ShanghaiTech University, Shanghai, China [2]University of Illinois at Urbana-Champaign, Illinois, United States. Correspondence to: Kan Ren <renkan@shanghaitech.edu.cn>.

*Proceedings of the 42$^{nd}$ International Conference on Machine Learning*, Vancouver, Canada. PMLR 267, 2025. Copyright 2025 by the author(s).

pressions compared to traditional structured conditions, as illustrated in Fig. 1. By leveraging text descriptions for time series generation, our approach enables fine-grained control and communicates richer semantic information, offering users enhanced flexibility and control over the generation.

This novel task presents several challenges stemming from the target of multi-modality modeling and the unique characteristics of time series, which cannot be effectively addressed using image generation approaches (Song et al., 2021; Peebles & Xie, 2023). First, unstructured data, such as text, often carries intricate semantic information, making it difficult to distinguish relevant content from irrelevant noise (Yin et al., 2019). Additionally, time series data exhibit multivariate characteristics and complex temporal dependencies (Torres et al., 2021; Wu et al., 2021), which fundamentally differ from the spatial structures typically encountered in image generation tasks. Furthermore, the control conditions for time series generation encompass multi-semantic information and exert varying influences. For example, textual conditions like "raining the whole day" and "raining after sunny weather" introduce nuanced distinctions that must be accurately reflected in the generated time series. Finally, the scarcity of paired time series and text data in real-world scenarios hinders the learning of robust connections between the two modalities.

To tackle these challenges, we propose VERBALTS, a novel multi-focal alignment and generation framework based on vanilla diffusion models (Ho et al., 2020) for text-to-time-series generation. We first recognize that time series exhibit multi-semantic information, reflected through representations and generations at different resolutions. VERBALTS captures both the hierarchical semantics in text conditions and their multi-resolution influences on the generated time series. Specifically, low-resolution representations capture global trends, while high-resolution representations focus on finer details, such as local shapelets. Second, considering that diffusion models process information and generate content at varying granularities (Zhang et al., 2024a; Fan et al., 2024), VERBALTS segments the generation process into multiple stages, applying condition information dynamically. Additionally, our framework extends to multivariate time series generation with nuanced semantic information alignment. Beyond traditional diffusion models (Song et al., 2021), our proposed method VERBALTS integrates a multi-view time series noise estimator with a multi-focal text processor, capturing the nuanced interplay between textual conditions and time series dynamics.

Our main contributions are summarized as follows. (i) We introduce the novel task of time series generation from unstructured data, to the best of our knowledge, marking the first work to address this challenge and enabling more fine-grained controlled generation. (ii) To effectively model the complex relationships between text and time series, we incorporate an innovative multi-focal alignment and generation framework. (iii) We construct a comprehensive benchmark with multi-facet time series datasets and textual information, which evaluates both generation and semantic editing performance of different methods.

## 2. Related Work

### 2.1. Conditional Time Series Generation

Conditional time series generation has attracted significant attention in recent years (Ang et al., 2023; Yang et al., 2024; Zhang et al., 2024c). Early approaches primarily relied on generative adversarial networks (GANs) (Esteban et al., 2017; Wang et al., 2023; Li et al., 2022) where a generator and a discriminator were jointly trained to refine generation quality through adversarial learning. Variational autoencoders (VAEs) (Lee et al., 2023) were also explored, optimizing the evidence lower bound (ELBO) to approximate the underlying data distributions. More recently, diffusion models (Coletta et al., 2024; Narasimhan et al., 2024; Tashiro et al., 2021) have emerged as a more stable and reliable alternative, consistently demonstrating superior performance in time series synthesis.

From the perspective of control conditions, time series generation can be categorized into structured and unstructured conditions. Structured conditions follow a fixed format, such as discrete categories (Li et al., 2022; Lee et al., 2023), metadata (Narasimhan et al., 2024) or preset constraints (Coletta et al., 2024). While easy to learn, these approaches are inherently constrained by the predefined condition space, restricting the diversity of generated samples. For instance, DiffTime (Coletta et al., 2024) incorporates constraints as conditions, struggling with conditions that cannot be easily formulated as constraints, ultimately limiting its generalizability. In contrast, unstructured conditions offer greater flexibility. Several works (Chen et al.; Chung et al., 2023) attempt to utilize clinical text reports to generate electrocardiograph data. However, these methods rely on domain knowledge, which are largely domain-specific and lack the generalization capability required for broader time series applications.

### 2.2. Time Series with Multi-modal Data

Prior studies (Liu et al.; Wang et al., 2024c) suggest that numerical time series alone often lack sufficient information, while external multi-modal data can provide valuable complementary insights. This has sparked growing interest in jointly modeling time series alongside other modalities. Some works (Han et al., 2024; Duan et al., 2023; Akbari et al., 2021) focus on aligning different modalities such as text, images, and time series to learn unified represen-

tations. Others (Xu et al., 2024; Srinivas et al., 2024; Jia et al., 2024; Xue & Salim, 2023; Liu et al., 2024) utilize the coarse-grained textual information to enhance time series forecasting. Additionally, some exploration like (Li et al., 2023) aims to describe time series in natural language, thereby improving their interpretability. Together, these efforts collectively highlight the feasibility and necessity of integrating multi-modal data to improve time series modeling. Building on this trend, we explore multi-modal time series generation, aiming to strengthen connections between time series and other modalities.

### 2.3. Multi-scale Time Series Modeling

The complex temporal dependencies inherent in time series present significant modeling challenges, motivating the development of multi-resolution and multi-scale approaches. These methods are particularly prominent in forecasting tasks. For example, TimeMixer (Wang et al., 2024b) and TimeMixer++ (Wang et al., 2024a) decompose time series into trend and seasonality components, employing top-down and bottom-up mixing strategies at different resolutions, respectively. Pathformer (Chen et al., 2024a) introduces a multi-scale routing mechanism for dynamically selecting appropriate transformers. MG-TSD (Fan et al., 2024) models the denoising process in diffusion as a progressive refinement from low to high resolution, using downsampled series as supervision signals at various diffusion steps. In time series editing, TEdit (Jing et al., 2024a) operates on variable patch sizes to enable parallel multi-resolution editing. For representation learning, TS2Vec (Yue et al., 2022) performs hierarchical contrastive learning to obtain robust contextual representations. However, these methods primarily focus on intra-modal multi-scale structures, overlooking the hierarchical information present in other modalities and the cross-modal interactions that are crucial for effective multi-modal modeling.

## 3. Text To Time Series Generation

### 3.1. Problem Formulation

Consider a sample pair of time series $\mathbf{x}$ and its corresponding textual description $\mathbf{c}$. Given a text description $\mathbf{c} \in \mathbb{N}^M$ with $M$ tokens, we aim to learn a generative model $f$, to generate a time series that conforms to the text $\hat{\mathbf{x}} = f(\mathbf{c}) \in \mathbb{R}^{K \times L}$, where $K$ and $L$ are the number of variables and the length, respectively.

### 3.2. Conditional Diffusion Model Framework

Our model is built upon the conditional diffusion model (Saharia et al., 2022), We briefly review the training and inference procedures below.

During training, noise is gradually added to the original data distribution $q(\mathbf{x}_0)$[1] via a Gaussian Markov transition:

$$q(\mathbf{x}_{1:T}|\mathbf{x}_0) = \prod_{t=1}^{T} q(\mathbf{x}_t|\mathbf{x}_{t-1}),$$
$$q(\mathbf{x}_t|\mathbf{x}_{t-1}) = \mathcal{N}(\mathbf{x}_{t-1}; \sqrt{1-\beta_t}\mathbf{x}_{t-1}, \beta_t\mathbf{I}), \quad (1)$$

and produces the noisy sample $\mathbf{x}_t$ at each diffusion step $t \in [1, T]$. Here $\{\beta_t\}_{t=1}^{T}$ are the predetermined variance schedule. $\mathbf{x}_t$ can be expressed as $\mathbf{x}_t = \sqrt{\alpha_t}\mathbf{x}_0 + \sqrt{1-\alpha_t}\boldsymbol{\epsilon}$, where $\boldsymbol{\epsilon} \sim \mathcal{N}(\mathbf{0}, \mathbf{I})$ and $\alpha_t := \prod_{s=1}^{t}(1-\beta_s)$.

Then, a learnable noise estimation network $\boldsymbol{\epsilon}_\theta(\mathbf{x}_t, t, g_\phi(\mathbf{c}))$ with the condition processor $g_\phi$ are trained by estimating the noise added to $\mathbf{x}_t$, which is used in the reverse process of diffusion model described later. The objective function is to minimize the noise estimation loss as

$$\min_{\theta,\phi} \mathcal{L}(\mathbf{x}_0) = \min_{\theta,\phi} \mathbb{E}_{\boldsymbol{\epsilon}\sim\mathcal{N}(\mathbf{0},\mathbf{I}),t\sim\mathcal{U}(1,T)} \|\boldsymbol{\epsilon} - \boldsymbol{\epsilon}_\theta(\mathbf{x}_t, t, g_\phi(\mathbf{c}))\|_2^2 . \quad (2)$$

$g_\phi(\mathbf{c})$ denotes the encoder of the condition $\mathbf{c}$ and will be further discussed in Sec. 4.2, $\mathbf{x}_0 \sim q(\mathbf{x}_0)$ is sampled from the real data distribution, $\mathbf{x}_t$ is a noisy version of $\mathbf{x}_0$.

During inference, given a condition $\mathbf{c}$ processed by $g_\phi(\mathbf{c})$, a sample $\hat{\mathbf{x}}_0$ can be generated from random noise $\hat{\mathbf{x}}_T \sim \mathcal{N}(\mathbf{0}, \mathbf{I})$ through multiple steps of denoising which uses the trained noise estimator $\boldsymbol{\epsilon}_\theta(\hat{\mathbf{x}}_t, t, g_\phi(\mathbf{c}))$ with a sampler, such as the deterministic Denoising Diffusion Implicit Model (DDIM) (Song et al., 2021)

$$\hat{\mathbf{x}}_{t-1} = \frac{\sqrt{\alpha_{t-1}}}{\sqrt{\alpha_t}}(\hat{\mathbf{x}}_t - \sqrt{1-\alpha_t}\boldsymbol{\epsilon}_\theta(\hat{\mathbf{x}}_t, t, g_\phi(\mathbf{c}))) + \sqrt{1-\alpha_{t-1}}\boldsymbol{\epsilon}_\theta(\hat{\mathbf{x}}_t, t, g_\phi(\mathbf{c})). \quad (3)$$

Consequently, it is critical for the noise estimator in time series generation to effectively capture the spatio-temporal properties while accounting for the nuanced influences of textual information on the generation process. A significant challenge lies in bridging the gap between unstructured textual descriptions and the time series modality. Aligning the generation of time series with the multi-semantic information embedded in text is a non-trivial task, as illustrated in Sec. 1. In the following section, we will introduce our proposed method, VERBALTS.

## 4. VERBALTS: Multi-focal Alignment and Generation

In this section, we introduce our method, VERBALTS, which consists of two key components: a multi-view noise estimator $\boldsymbol{\epsilon}_\theta$ and a multi-focal text processor $g_\phi$. The noise estimator $\boldsymbol{\epsilon}_\theta$ is designed to model and generate time series

---

[1] $\mathbf{x}$ and $\mathbf{x}_0$ are interchangeable in this paper.

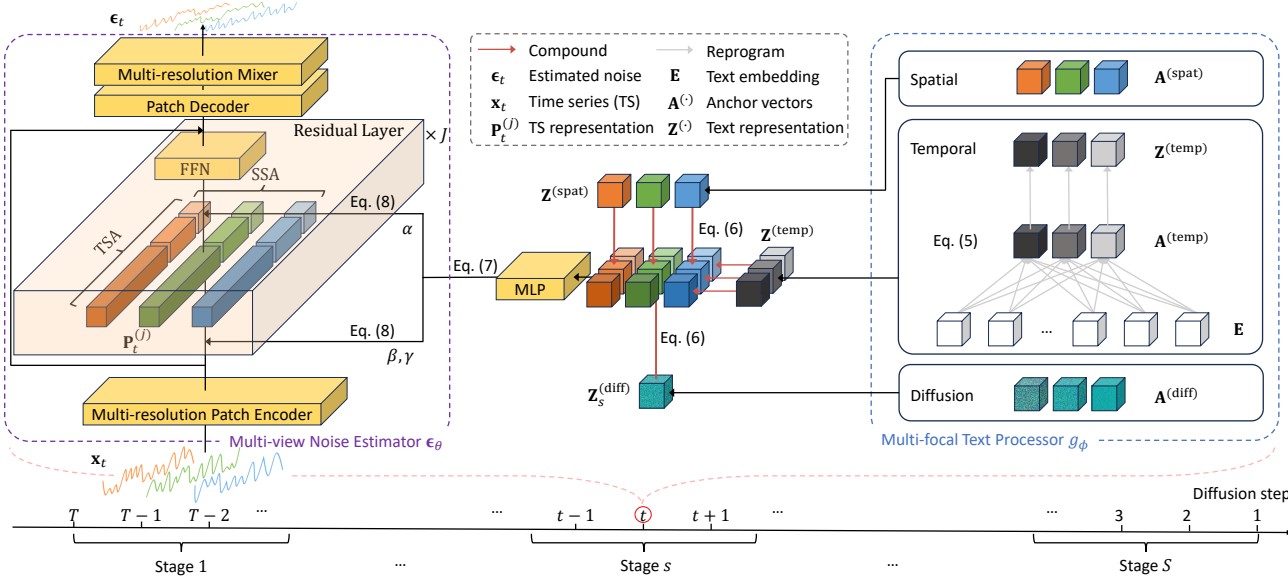

*Figure 2.* The overall model architecture of VERBALTS, including multi-view noise estimator and multi-focal text processor.

with finer granularity, effectively capturing detailed temporal dynamics, incorporating spatial interactions among variables, and ensuring nuanced condition control throughout the diffusion process (Sec. 4.1). Meanwhile, the text processor $g_\phi$ transforms textual information into a multi-semantic structured latent space, enabling a more precise representation of the complex relationships between text and time series modalities (Sec. 4.2). Finally, the multi-semantic information would be aligned to the time series generation process, with an elaborately designed conditional control mechanism (Sec. 4.3). The overall model architecture is presented in Fig. 2.

### 4.1. Multi-view Noise Estimator

Textual conditions provide the flexible descriptions of the target time series from various perspectives, as mentioned in Sec. 1. Generating time series from these flexible and complex textual conditions requires fine-grained control over the generation process. In this paper, we derive the fine-grained time series generation from the three key views of the generation process. First, from the **temporal view**, time series data exhibit strong sequential characteristics. Jing et al. (2024a) has demonstrated that the influence of control conditions on time series generation varies across different time spans, making finer-granularity temporal modeling essential. Second, from the **spatial view**, real-world time series data usually contains multiple variables, yet most existing studies focus on univariate time series generation (Jing et al., 2024a), overlooking variable-level interactions. Furthermore, from the **diffusion view**, recent studies (Zhang et al., 2024a; Fan et al., 2024) have discovered that different steps in the denoising process in the diffusion model derive the generation details at varying levels of granularity. In

the following content, we first provide an overview of the proposed multi-view noise estimator and then elaborate on each individual view.

**Overview.** As demonstrated in Fig. 2, given a noisy time series $\mathbf{x}_t$, the noise estimator $\boldsymbol{\epsilon}_\theta(\mathbf{x}_t, t, \cdot)$ first encode it into embedding space via the patch encoder, which will be passed through $J$ residual layers and finally decoded by the patch decoder and multi-resolution mixer to obtain the estimated noise $\boldsymbol{\epsilon}_t$, that will be removed from $\mathbf{x}_t$. Each residual layer contains two blocks: temporal self-attention TSA$(\cdot)$, spatial self-attention SSA$(\cdot)$ (Vaswani, 2017), with a feed-forward network FFN$(\cdot)$. Please refer to Appendix B for more details about the model architecture.

**Temporal View.** The input noisy time series $\mathbf{x}_t \in \mathbb{R}^{K \times L}$ is modeled with a total of $R$ resolutions in the temporal dimension. First, at each resolution $r$, $\mathbf{x}_t$ is sliced into $N_r = \lfloor \frac{L - L_r}{L_r} \rfloor$ patches of size $L_r$ to get $\mathbf{x}_{t,r} \in \mathbb{R}^{K \times N_r \times L_r}$. $\mathbf{x}_{t,r}$ is then encoded into embeddings $\mathbf{p}_{t,r} \in \mathbb{R}^{K \times N_r \times D}$ via a linear layer, where $D$ is the size of the embedding. Then we concatenate the $N = \Sigma_{r=1}^R N_r$ patch embeddings from $R$ resolutions into a single tensor $\mathbf{P}_t \in \mathbb{R}^{K \times N \times D}$, which will be fed into $J$ residual layers. Within the $j$-th residual layer, we design a TSA$(\cdot)$ block to capture multi-resolution temporal information of $\mathbf{P}_t^{(j-1)}$ by performing the self-attention (Vaswani, 2017) on the temporal dimension. Note that we use a mask matrix to restrict attention operations within the same resolution. More details can be referred to Appendix B.1.

**Spatial View.** Within the $j$-th layer, we design a SSA$(\cdot)$ block, similar to TSA$(\cdot)$, to capture the spatial information by applying the self-attention over the $K$ variables in the spatial dimension of $\mathbf{P}_t^{(j-1)}$.

The complete process of temporal and spatial modeling in the $j$-th layer of the noise estimator can be expressed as

$$\mathbf{P}_t^{(j)} = \mathbf{P}_t^{(j-1)} + \text{FFN}(\text{SSA}(\text{TSA}(\mathbf{P}_t^{(j-1)} + \mathbf{e}_t))), \quad (4)$$

where $\mathbf{P}_t^{(0)} = \mathbf{P}_t$, $\mathbf{e}_t \in \mathbb{R}^D$ is the embedding of the current denoising time step, with spatial and temporal self-attention mechanism mentioned above. More details are presented in Appendix B.1.

**Diffusion View.** Now we move attention onto the overall diffusion process $\boldsymbol{\epsilon}_\theta(\mathbf{x}_t, t, \cdot)$, $t \in [1, T]$. We evenly divide the total $T$ denoising steps into $S$ stages. For the denoising step $t$, it is assigned to the $s$-th stage where $s = \lfloor \frac{t \times S}{T} \rfloor$. At each stage, the diffusion model generates time series at different granularities, guided by corresponding levels of textual information, as detailed in Sec. 4.3. For clarity, we represent the multi-view time series representations across the entire diffusion process as a unified variable $\bar{\mathbf{P}} \in \mathbb{R}^{T \times K \times N \times D}$ concatenating the temporal and spatial representation $\mathbf{P}_t$ for each diffusion step $t \in [1, T]$.

Till now, the proposed multi-view time series modeling and generation framework in the noise estimator has been clarified. It is worth noticing that, the control of textual information on the three views, i.e., different temporal resolutions, spatial variables and multiple diffusion stages, would be further described in Sec. 4.3. Unlike existing time series diffusion models (Tashiro et al., 2021; Narasimhan et al., 2024; Coletta et al., 2024), our approach models the generation of time series from three views: spatiotemporal relationships and the noise reduction process. This design, assembly with the multi-focal alignment module described in Sec. 4.3, enables more nuanced generation and leads to higher-quality outputs, as is shown in the Sec. 5.2.

### 4.2. Multi-focal Text Processor

Textual conditions encode complex multi-semantic information within an unstructured token sequence. The effectiveness of the noise estimator $\boldsymbol{\epsilon}_\theta$ is strongly influenced by how well the textual conditions are modeled. However, modeling this information presents several challenges. First, aligning time series components with relevant textual semantics is non-trivial.. From the temporal view, phrases like "at the beginning" and "around the middle" describe local features, while "overall" and "global" refer to broader trends. From the spatial view, terms like "light rain" primarily affect the variable of precipitation while having minimal influence on the variable of wind direction. From the diffusion view, tokens conveying global information are likely to play a larger role in early diffusion stages, whereas tokens with fine-grained details become more relevant in later stages. Second, textual data includes grammatical fillers (e.g., "the", "is", and "of") that carry little semantic weight and should not be overemphasized during modeling.

To address these challenges, we propose a multi-focal text processor, denoted as $g_\phi(\mathbf{c})$, for better information alignment. This text processor transforms a complete text sentence into multi-semantic components from the views of temporal, spatial dimensions and diffusion process, through semantic reprogramming (Chen, 2024), as introduced as below. We term it "multi-focal" because textual semantics are distributed across different tokens, and our approach selectively focuses on the most relevant ones.

**Multi-focal Reprogramming**. Given a text condition $\mathbf{c} \in \mathbb{N}^M$ of $M$ tokens, we first encode $\mathbf{c}$ into embeddings $\mathbf{E} \in \mathbb{R}^{M \times D}$ via a pre-trained text encoder, e.g., Long-clip (Zhang et al., 2024b). Then, we reprogram $\mathbf{E}$ into $I$ semantic levels $\mathbf{Z} \in \mathbb{R}^{I \times D}$ based on the learnable anchor vectors $\mathbf{A} \in \mathbb{R}^{I \times D}$, where the $i$-th row corresponds to the $i$-th semantic level. Specifically, the reprogram (RPG) operation is defined by:

$$\mathbf{Z} = \text{RPG}(\mathbf{Q}_A, \mathbf{K}_E, \mathbf{V}_E) = \text{Softmax}(\frac{\mathbf{Q}_A \mathbf{K}_E^\top}{\sqrt{D}})\mathbf{V}_E, \quad (5)$$

where $\mathbf{Q}_A = \mathbf{A}\mathbf{W}_Q$, $\mathbf{K}_E = \mathbf{E}\mathbf{W}_K$, $\mathbf{V}_E = \mathbf{E}\mathbf{W}_V$, are query, key, and value matrices, respectively, and $\mathbf{W}_{(\cdot)}$ are learnable weights.

From the three views of multi-focal modeling, we specifically define: (i) Diffusion semantic anchors $\mathbf{A}^{(\text{diff})} \in \mathbb{R}^{S \times D}$ where $S$ is the stage number of denoising process; (ii) Temporal semantic anchors $\mathbf{A}^{(\text{temp})} \in \mathbb{R}^{R \times D}$ where $R$ is the resolution number of time series modeling; (iii) Spatial semantic anchors $\mathbf{A}^{(\text{spat})} \in \mathbb{R}^{K \times D}$ where $K$ is the variable number. The corresponding text representations can be calculated by the reprogramming mechanism, resulting in $\mathbf{Z}^{(\text{diff})} \in \mathbb{R}^{S \times 1 \times 1 \times D}$, $\mathbf{Z}^{(\text{temp})} \in \mathbb{R}^{1 \times 1 \times R \times D}$, $\mathbf{Z}^{(\text{spat})} \in \mathbb{R}^{1 \times K \times 1 \times D}$, respectively. By aligning multi-semantic conditions with the corresponding time series components (Sec. 4.3) and updating them through gradient backpropagation, distinct anchor vectors learn unique representations, as demonstrated in Sec. 5.4.3.

Further, we unify the multi-semantic representations of the temporal, spatial and diffusion process views to a compound matrix $\bar{\mathbf{Z}} \in \mathbb{R}^{S \times K \times R \times D}$ as below.

$$\bar{\mathbf{Z}} := \mathbf{Z}^{(\text{diff})} \circledast \mathbf{Z}^{(\text{spat})} \circledast \mathbf{Z}^{(\text{temp})}, \text{where}$$
$$\bar{\mathbf{Z}}_{s,k,r} = \mathbf{Z}_s^{(\text{diff})} + \mathbf{Z}_k^{(\text{spat})} + \mathbf{Z}_r^{(\text{temp})} . \quad (6)$$

The $\circledast$ is the operation of matrix addition with broadcasting. The representation $\bar{\mathbf{Z}}$ not only encodes various multi-semantic textual information but also enables multi-modal control for time series generation, which will be further discussed in Sec. 4.3.

Unlike the text processor in visual generation (Rombach et al., 2022), which uses a unified text condition throughout, our multi-focal text processor leverages hierarchical textual

information. This design facilitates multi-semantic pattern discovery and enables more refined generation, as demonstrated in Secs. 5.3 and 5.4. Furthermore, we introduce the learnable anchor vectors to eliminate the need for manual extraction of structured information from unstructured text, enhancing both flexibility and efficiency and exhibiting more superior performance than the strong baselines based on structured conditions, as shown in Sec. 5.2.

## 4.3. Multi-modality Semantic Alignment

Using the multi-view noise estimator and multi-focal text processor described earlier, we obtain multi-view representations for the time series, $\bar{\mathbf{P}} \in \mathbb{R}^{T \times K \times N \times D}$, and for the text, $\bar{\mathbf{Z}} \in \mathbb{R}^{S \times K \times R \times D}$. The key challenge now lies in effectively aligning and applying the semantic information from the text descriptions into the diffusion denoising process.

To address this, we propose an adapter that aligns the multi-semantic information of the text across its three views with the corresponding components of the time series, ensuring seamless integration during the diffusion process. Inspired by the adaptive layer normalization (Xu et al., 2019), the proposed adapter rectifies the time series embedding $\bar{\mathbf{P}}$ based on the textual condition $\bar{\mathbf{Z}}$ via a gate $\alpha$, a scaler $\gamma$ and a shifter $\beta$, which are calculated through a Multi-Layer Perceptron (MLP):

$$\alpha, \beta, \gamma = \text{MLP}(\bar{\mathbf{Z}}) \text{ , where } \alpha, \beta, \gamma \in \mathbb{R}^{S \times K \times R \times D} . \quad (7)$$

Each dimension of the controlling parameters manages the influence on time series modeling along with $R$ resolutions and $K$ variables within the $S$ stages of the diffusion process correspondingly, as explained in Sec. 4.1. The representations within the diffusion model are controlled via

$$\tilde{\mathbf{P}} = \alpha \odot \text{SSA}(\text{TSA}(\gamma \odot (\bar{\mathbf{P}} + \mathbf{e}) \oplus \beta)), \quad (8)$$

$\odot$ and $\oplus$ are custom element-wise multiplication and addition between matrices. The adapter parameters $\alpha, \beta, \gamma$ undergo broadcasted expansion through temporal and diffusion segment replication, achieving dimensional compatibility with tensor $\bar{\mathbf{P}}$, with detailed calculations provided in Appendix B.3. The conditionally altered representation $\tilde{\mathbf{P}} \in \mathbb{R}^{T \times K \times N \times D}$ will be passed into FFN in each residual layer for further calculation.

With the proposed adapter, we align multi-semantic information across temporal, spatial, and diffusion views between the two modalities. Unlike Liu et al. (2022), which adjusts weights for decomposed conditions, our approach considers the correspondence between control conditions and generated data, a crucial factor for fine-grained conditional generation. The effectiveness is detailed in Sec. 5.

## 5. Experiment

In this section, we present the experimental settings and the corresponding results. Our analysis follows the below research questions (RQs). **RQ1**: Does the unstructured condition conveys more information than the structured one for time series generation? **RQ2**: How does the multi-focal generation mechanism work? **RQ3**: How does the proposed method VERBALTS build the alignment between text and time series? We have released all the reproducible code and benchmarking datasets at https://seqml.github.io/VerbalTS/.

### 5.1. Experiment Setup

**Datasets.** To solve the problem of scarcity of paired text and time series data, we construct datasets using data from three different sources: (i) Full synthetic datasets with manually constructed texts and the corresponding ground-truth time series, including **Synth-U** with univariate time series and **Synth-M** with multivariate time series. (ii) Real-world datasets including **Weather** (Xu et al., 2024) of climtime indicators and **BlindWays** (Kim et al., 2024) of blind people's trajectories, both originating from real-world textual annotations and time series. (iii) Augmented real-world datasets including **ETTm1** (Zhou et al., 2021) and **Traffic** (Leo, 2024) datasets with the real-world time series and the corresponding textual descriptions annotated by the external tool. For detail information of the dataset construction, please refer to Appendix A.

**Evaluation metrics.** We evaluate the quality of the generated time series from two perspectives. (i) Fidelity: Frechet Inception Distance (**FID**) (Heusel et al., 2017) and Joint Frechet Time Series Distance (**J-FTSD**) (Narasimhan et al., 2024) are used to evaluate the fidelity by measuring the discrepancy between the generated data distribution and the real data distribution. (ii) Semantic Alignment: Contrastive Time series Text Pretraing (**CTTP**) score is used to measure the semantic similarity between the generated time series $\hat{\mathbf{x}}$ and the text condition $\mathbf{c}$. Similar to CLIP score (Radford et al., 2021), a proxy model trained through contrastive learning calculates the similarity between the generated time series and the condition text in the latent space. We further provide the details of the model architecture of CTTP model and prove its reliability in Appendix C.

**Baselines.** Since there are few works studying the general unstructured text to time series generation problem, we compare our proposed method VERBALTS with attribute-based generative models **TimeWeaver** (Narasimhan et al., 2024) and **TEdit** (Jing et al., 2024a), constrained generation model **DiffTime** (Coletta et al., 2024) and class-conditioned generation model **TimeVQVAE** (Lee et al., 2023). The conditions for these methods are transformed from textual descriptions. More implementation details are provided in Appendix A.4.

| Multivariate setting | | Synthetic dataset | | | Real-world datasets | | | | | |
| --- | --- | --- | --- | --- | --- | --- | --- | --- | --- | --- |
| Condition | Method | ↓FID | Synth-M ↓JFTSD | ↑CTTP | ↓FID | BlindWays ↓JFTSD | ↑CTTP | ↓FID | Weather ↓JFTSD | ↑CTTP |
| Class | TimeVQVAE (Lee et al., 2023) | $83.17_{\pm0.13}$ | $94.16_{\pm0.10}$ | $51.23_{\pm0.22}$ | $29.48_{\pm3.30}$ | $36.87_{\pm3.20}$ | $8.63_{\pm0.09}$ | $54.26_{\pm0.30}$ | $59.46_{\pm0.25}$ | $22.19_{\pm0.10}$ |
| Constraint | DiffTime (Coletta et al., 2024) | $49.87_{\pm1.12}$ | $96.86_{\pm1.40}$ | $21.60_{\pm0.40}$ | $58.32_{\pm29.41}$ | $66.21_{\pm29.37}$ | $7.14_{\pm0.73}$ | $50.90_{\pm0.97}$ | $68.88_{\pm0.51}$ | $14.48_{\pm0.17}$ |
| Attribute | TimeWeaver (Narasimhan et al., 2024) | $43.38_{\pm0.00}$ | $59.59_{\pm0.00}$ | $52.15_{\pm0.00}$ | $51.31_{\pm26.27}$ | $59.22_{\pm26.29}$ | $7.50_{\pm0.48}$ | $16.07_{\pm0.11}$ | $19.79_{\pm0.20}$ | $27.27_{\pm0.13}$ |
| | TEdit (Jing et al., 2024a) | $41.94_{\pm1.92}$ | $58.08_{\pm1.59}$ | $52.55_{\pm0.20}$ | $28.96_{\pm2.96}$ | $36.73_{\pm2.91}$ | $7.96_{\pm0.41}$ | $14.86_{\pm0.31}$ | $18.33_{\pm0.30}$ | $27.56_{\pm0.07}$ |
| Text | VERBALTS (ours) | $\mathbf{29.05}_{\pm0.62}$ | $\mathbf{32.91}_{\pm0.62}$ | $64.13_{\pm0.06}$ | $\mathbf{27.63}_{\pm1.49}$ | $\mathbf{34.40}_{\pm1.55}$ | $9.18_{\pm0.26}$ | $\mathbf{6.13}_{\pm0.21}$ | $\mathbf{8.56}_{\pm0.21}$ | $30.62_{\pm0.03}$ |

| Univariate setting | | Synthetic dataset | | | Real-world datasets | | | | | |
| --- | --- | --- | --- | --- | --- | --- | --- | --- | --- | --- |
| Condition | Method | ↓FID | Synth-U ↓JFTSD | ↑CTTP | ↓FID | ETTm1 ↓JFTSD | ↑CTTP | ↓FID | Traffic ↓JFTSD | ↑CTTP |
| Class | TimeVQVAE (Lee et al., 2023) | $41.29_{\pm1.23}$ | $54.47_{\pm1.01}$ | $26.94_{\pm0.07}$ | $26.84_{\pm0.91}$ | $30.19_{\pm0.94}$ | $10.56_{\pm0.16}$ | $40.47_{\pm0.57}$ | $42.68_{\pm0.52}$ | $\mathbf{8.52}_{\pm0.08}$ |
| Constraint | DiffTime (Coletta et al., 2024) | $45.42_{\pm1.72}$ | $72.94_{\pm1.53}$ | $12.79_{\pm0.12}$ | $59.14_{\pm1.75}$ | $66.37_{\pm1.66}$ | $2.33_{\pm0.10}$ | $100.60_{\pm3.76}$ | $104.27_{\pm3.75}$ | $3.89_{\pm0.07}$ |
| Attribute | TimeWeaver (Narasimhan et al., 2024) | $36.38_{\pm1.26}$ | $49.96_{\pm1.07}$ | $27.95_{\pm0.08}$ | $32.90_{\pm0.55}$ | $36.26_{\pm0.51}$ | $10.34_{\pm0.05}$ | $44.34_{\pm1.69}$ | $46.66_{\pm1.67}$ | $8.31_{\pm0.05}$ |
| | TEdit (Jing et al., 2024a) | $36.52_{\pm0.57}$ | $49.92_{\pm0.48}$ | $28.09_{\pm0.06}$ | $33.04_{\pm0.45}$ | $36.38_{\pm0.46}$ | $10.34_{\pm0.05}$ | $45.82_{\pm1.83}$ | $48.19_{\pm1.82}$ | $8.20_{\pm0.03}$ |
| Text | VERBALTS (ours) | $\mathbf{28.29}_{\pm0.29}$ | $\mathbf{32.26}_{\pm0.23}$ | $37.56_{\pm0.07}$ | $\mathbf{24.19}_{\pm0.53}$ | $\mathbf{27.10}_{\pm0.58}$ | $11.10_{\pm0.10}$ | $37.03_{\pm1.53}$ | $39.03_{\pm1.54}$ | $8.43_{\pm0.03}$ |

*Table 1.* Averaged performance with standard deviation (mean±std) of multivariate and univariate settings on synthetic (left) and real-world (right) datasets with three random runs. The best performance is with bold font, ↑ (↓) means the higher (lower), the better.

| Dataset | Metric | VERBALTS | w/o $\mathbf{A}^s$ | w/o $\mathbf{A}^s$ & $\mathbf{A}^t$ | w/o $\mathbf{A}^s$ & $\mathbf{A}^t$ & $\mathbf{A}^d$ |
| --- | --- | --- | --- | --- | --- |
| Synth-M | ↓FID | $\mathbf{29.05}_{\pm0.62}$ | $29.98_{\pm0.68}$ | $34.17_{\pm0.56}$ | $34.18_{\pm0.39}$ |
| | ↓JFTSD | $\mathbf{32.91}_{\pm0.62}$ | $33.91_{\pm0.65}$ | $40.95_{\pm0.51}$ | $41.43_{\pm0.61}$ |
| | ↑CTTP | $\mathbf{64.13}_{\pm0.06}$ | $63.86_{\pm0.04}$ | $60.17_{\pm0.39}$ | $60.07_{\pm0.26}$ |
| Weather | ↓FID | $\mathbf{6.13}_{\pm0.21}$ | $7.01_{\pm0.33}$ | $7.30_{\pm0.25}$ | $7.60_{\pm0.41}$ |
| | ↓JFTSD | $\mathbf{8.56}_{\pm0.21}$ | $9.43_{\pm0.32}$ | $9.76_{\pm0.28}$ | $10.12_{\pm0.38}$ |
| | ↑CTTP | $\mathbf{30.62}_{\pm0.03}$ | $30.32_{\pm0.06}$ | $30.23_{\pm0.08}$ | $29.95_{\pm0.07}$ |

*Table 2.* Ablation studies on Synth-M and Weather datasets. $\mathbf{A}^s$, $\mathbf{A}^t$ and $\mathbf{A}^d$ are the abbreviations of $\mathbf{A}^{(\text{spat})}$, $\mathbf{A}^{(\text{temp})}$ and $\mathbf{A}^{(\text{diff})}$, which refer to the anchor vectors of spatial, temporal and diffusion in the multi-focal text processor, respectively.

## 5.2. Quantitative Results

In this section, we quantitatively evaluate and report the performance of the compared methods for time series generation from given textual conditions (**RQ1**) across all datasets. The average results for multivariate and univariate time series generation are presented in Tab. 1.

As shown in Tab. 1, VERBALTS demonstrates superior fidelity on both synthetic and real-world datasets, achieving over 20% improvements on the Synth-M/U and Weather datasets. Moreover, VERBALTS achieves significantly better semantic alignment between the generated time series and the descriptive texts, as evidenced by its much higher CTTP performance. These results highlight not only the richer information embedded in unstructured text conditions but also the exceptional ability of our proposed method to generate high-quality time series that are semantically well-aligned with the given conditions. More qualitative evaluation results are provided in Appendix F.2.

## 5.3. Ablation Study

We conduct an ablation study of VERBALTS on the Synth-M and Weather datasets to evaluate the impact of the proposed multi-focal text processor and multi-view noise estimator, as described in Sec. 4.2 and 4.1 (**RQ2**).

In this study, we successively ablate the multi-focal modeling and generation capabilities of VERBALTS across the spa-

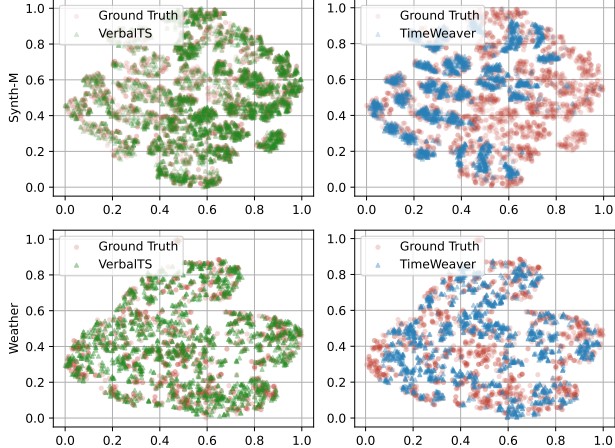

*Figure 3.* Comparison of generated data distribution between the VERBALTS (left) and TimeWeaver (right) on Synth-M and Weather datasets.

tial, temporal, and diffusion views by sequentially removing the anchor vectors $\mathbf{A}^{(\text{spat})}$, $\mathbf{A}^{(\text{temp})}$, and $\mathbf{A}^{(\text{diff})}$. Correspondingly, we simplify the calculation of the text representation $\hat{\mathbf{Z}} \in \mathbb{R}^{1\times1\times1\times D}$ by averaging the text sequence embedding $\mathbf{E}$ along the text length dimension. As shown in Tab. 2, each ablation operation results in a significant decline in performance, highlighting the positive effects of multi-focal modeling in VERBALTS for generating time series data that are both high-fidelity and semantically well-aligned with the textual conditions.

## 5.4. Extended Analysis

### 5.4.1. UNSTRUCTURED V.S. STRUCTURED CONDITION

We conduct analytical experiments, trying to uncover the effects of the unstructured text conditions versus the structured condition on time series generation (**RQ1**). We also analyze the behavior patterns of different compared models on two condition types.

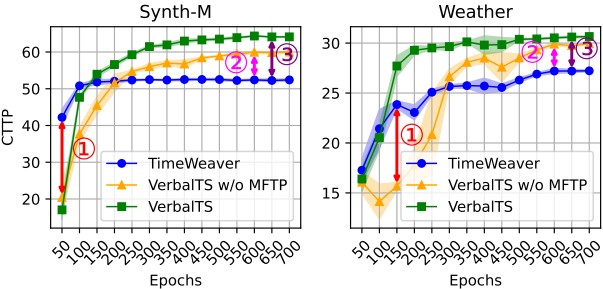

*Figure 4.* The learning curves of CTTP score on the validation set of Synth-M and Weather datasets. Findings: ①: Structured conditions are easier for model to learn. ②: Unstructured conditions provide additional information. ③: The multi-focal text processor (MFTP) in VERBALTS makes learning more effective.

**Finding 1**: *Time series generated from unstructured text conditions are more diverse and better aligned with the ground-truth distribution than those from structured conditions.* As shown in Fig. 3, t-SNE (Van der Maaten & Hinton, 2008) visualization reveals that text-based conditions produce more diverse samples, while structured conditions lead to stronger clustering and less variety. This highlights the limitations of structured conditions and the value of flexible controls like text for generating realistic time series. For more visualization results, please refer to Appendix F.1.

**Finding 2**: *Unstructured text conditions convey richer information but introduce additional noise, while this can be effectively handled by the multi-focal mechanism of* VERBALTS. We present the learning curves on the Synth-M and Weather datasets in Fig. 4. Initially, TimeWeaver exhibits faster learning (gap ①), as the structured conditions are noise-free, allowing the model to more easily extract useful information. However, it is later outperformed by VERBALTS (gap ②), owing to the richer, fine-grained details provided by the unstructured text. Furthermore, the multi-focal mechanism in VERBALTS enables more effective utilization of this additional information, leading to faster convergence and ultimately superior performance in text-to-time series generation (gap ③).

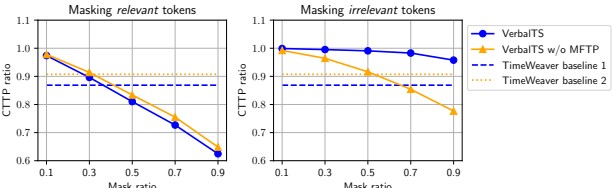

*Figure 5.* Intervention experiment on Synth-M dataset. We compare VERBALTS with and without the multi-focal text processor (MFTP). TimeWeaver baseline 1 and 2 serve as references without masking, representing the CTTP ratio of TimeWeaver to VERBALTS and VERBALTS w/o MFTP, respectively.

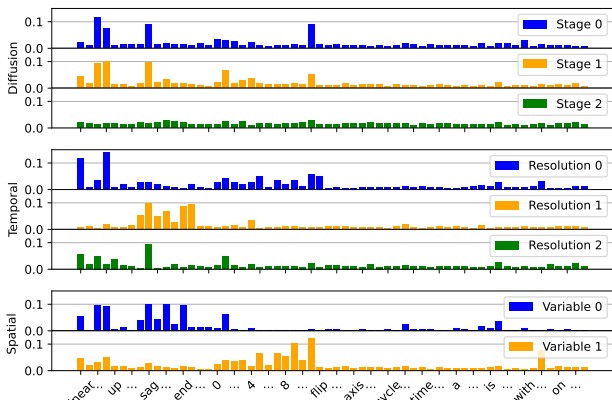

*Figure 6.* Distribution of the averaged attention weight in Eq. (5) of each token within the vocabulary. VERBALTS exhibits various attention distributions on different textual information from different views, showcasing the effect of multi-focal processing.

### 5.4.2. TEXT EFFECT IN TIME SERIES GENERATION

We further explore how text information impacts time series generation in VERBALTS, addressing **RQ3**.

**Finding 3**: VERBALTS *focuses on semantically relevant information in raw text, with the multi-focal text processor enhancing this ability.* To analyze the impact of relevant and irrelevant information, we manually labeled the token vocabulary into a relevant token set $\mathcal{V}^{(\mathrm{rel})}$ (e.g., descriptions of trend, shape, and seasonality) and an irrelevant token set $\mathcal{V}^{(\mathrm{irr})}$ (e.g., meaningless pronouns and stop words). More details are in Appendix A.1.5. We then conducted an intervention experiment by randomly masking relevant or irrelevant tokens and evaluating the effect on time series generation performance. Fig. 5 shows the quantitative analysis of the CTTP performance ratio $\frac{\mathrm{CTTP_{mask}}}{\mathrm{CTTP}}$ after masking tokens at varying ratios. The results demonstrate that masking semantically relevant tokens significantly degrades the generation performance (left of Fig. 5), whereas masking irrelevant tokens has minimal impact (right of Fig. 5). This resilience is attributed to the multi-focal text processor and alignment mechanisms in VERBALTS. Moreover, VERBALTS performs much better than baseline TimeWeaver even masking 90% irrelevant tokens in text, highlighting the robustness of our method. We further show a case study in Appendix F.3, to qualitatively illustrate how our method works.

### 5.4.3. EFFECT OF MULTI-FOCAL TEXT PROCESSING

**Finding 4**: *The multi-focal text processor enables* VERBALTS *to differentiate and utilize diverse textual information for fine-grained time series generation.* To evaluate how VERBALTS processes nuanced textual details (**RQ2**), we analyze the averaged attention values of each token in the vocabulary from Eq. (5). (See Appendix E.2 for more details.) As shown in Fig. 6, VERBALTS assigns varying

attention to tokens across different focuses, demonstrating the fine-grained modeling capabilities of the multi-focal text processor, as outlined in Sec. 4.2.

### 5.4.4. SENSITIVITY STUDY

**Finding 5**: VERBALTS *demonstrates robustness to hyperparameter settings. Beyond a certain threshold, the benefits of multi-resolution and multi-stage modeling become evident.* To evaluate the sensitivity of VERBALTS to specific hyperparameters, we conduct a sensitivity study on Synth-M and Weather datasets to evaluate the impact of the multiresolution number $R$ and multi-stage number $S$. As shown in Fig. 7, VERBALTS maintains stable performance across a wide range of hyperparameter values. Beyond a certain threshold, the benefits of incorporating multiple resolutions and stages become significant.

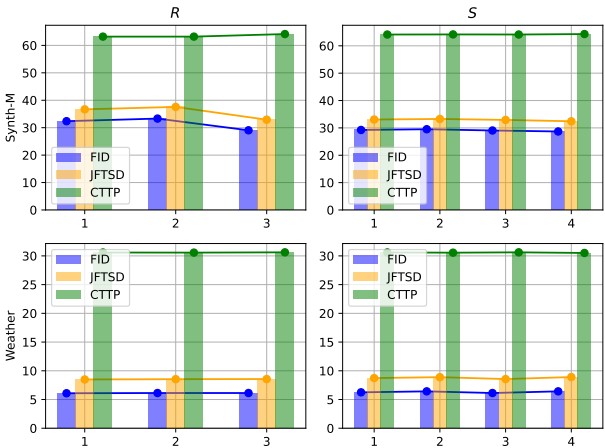

*Figure 7.* Sensitivity study on Synth-M and Weather dataset. $R$ is the resolution number and $S$ is the stage number of diffusion.

### 5.4.5. WIDER APPLICATION: TIME SERIES EDITING

**Finding 6**: VERBALTS *enables flexible and precise time series editing.* Adopting the approach from (Jing et al., 2024a), we integrate VERBALTS into the editing process. Specifically, noise is added to the source time series $\mathbf{x}^{\text{src}}$ via the forward process (Eq. (1)), and the reverse process (Eq. (3)) generates the edited time series $\hat{\mathbf{x}}^{\text{tgt}}$ under the target text condition via our proposed VERBALTS. Fig. 8 showcases the ability of VERBALTS to retain original text characteristics while aligning with modified descriptions. Compared to TEdit (Jing et al., 2024a), which relies on attributes, VERBALTS achieves finer edits by precisely locating and following the adjusted tokens in the target text description. More case study results are provided in Appendix F.4.

## 6. Conclusion

Text offers rich and nuanced information, making it a powerful modality for describing time series details or uncovering

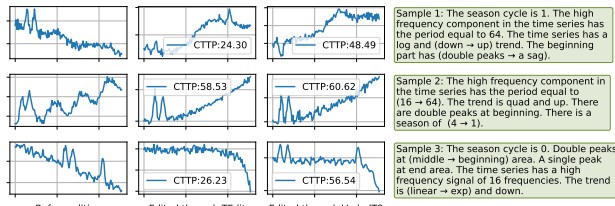

*Figure 8.* Illustration of editing task. Column 1: the raw time series before editing. Column 2: result edited by TEdit (Jing et al., 2024a); Column 3: result edited by our VERBALTS; Column 4: the condition prompts for editing, with (source → target) properties.

underlying mechanisms. In this work, we addressed the problem of generating time series from text by proposing VERBALTS, which integrates a novel multi-focal text processor with a multi-view diffusion model to achieve fine-grained semantic alignment between the two modalities. Experiments on two synthetic and four real-world datasets demonstrated the effectiveness of our approach through both quantitative and qualitative analyses. VERBALTS also showed strong capabilities in extracting meaningful text information, enabling high-quality generation and flexible applications such as controlled generation and editing.

However, our method has limitations, such as slower generation efficiency due to the long-range reverse process in diffusion models, which may hinder real-world applications. Addressing this limitation presents an important direction for future research.

## Impact Statement

This work introduces a novel method for generating time series from texts, bridging the gap between natural language understanding and time series synthesis. We foresee several potentially positive societal impacts: (i) providing richer time series data across various fields, (ii) reducing the risk of personal privacy leakage in data, and (iii) advancing the study of refined and controlled time series generation.

While we do not anticipate immediate or direct negative societal consequences arising from this contribution, we acknowledge that, like other generative technologies, it may be vulnerable to misuse. As such, responsible use, ethical oversight, and continued monitoring are essential to ensure that the technology is applied for beneficial purposes.

## Acknowledgment

The research was supported by National Natural Science Foundation of China (Grant No. 62406193). The authors also gratefully acknowledge further assistance provided by Shanghai Frontiers Science Center of Human-centered Artificial Intelligence, MoE Key Lab of Intelligent Perception and Human-Machine Collaboration, and HPC Platform of ShanghaiTech University.

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

| Attribute | Value |
|---|---|
| Trend Types | [Linear, Quadratic, Exponential, Logistic] |
| Trend Directions | [Up, Down] |
| Season Cycles | [0, 1, 2, 4] |
| Local Shapelets | [None, Single Peak, Sag, Double Peaks] |
| High Frequency Components | [0, 16, 32, 64] |
| Multivariable | [X-axis Flip, Y-axis Flip, Shift Forward, Shift Backward] |

*Table 3.* Summary of attribute options for synthetic datasets.

# A. Datasets

In this section, we provide detailed instructions for constructing the datasets, which include six datasets of three categories, as mentioned in Sec. 5.1.

## A.1. Full synthetic datasets

The Full synthetic represents the datasets that both time series and text description are manually generated, including the **Synth-U** and **Synth-M** datasets.

Specifically, we first manually define 6 types of attributes, as shown in Tab. 3. Based on these attributes, time series data are generated using specific mathematical formulas and multivariate rules; While textual descriptions are generated through predefined text templates. Thus, our processed dataset combines three types of data: generated time series, textual descriptions, and predefined attributes.

We generate Synth-U dataset according to the following formula:

$$\mathbf{x} = \mathbf{x}_{\text{trend}} + \mathbf{x}_{\text{season}} + \mathbf{x}_{\text{local}} + \mathbf{x}_{\text{hf}} + \mathbf{x}_{\text{noise}}$$

Among them, trend and season are the main components of time series, while noise, high-frequency components, and local shapelet are supplementary details. Therefore, we divide the control attributes into two categories: primary and secondary. The primary attributes are shared by all data in the data set, including **Trend Types**, **Trend Directions**, and **Season cycles**. The secondary attributes are attributes that are sample-specific and are difficult to be uniformly described, including **Local Shapelets** and **High Frequency Components**.

Synth-M follows a construction method similar to that of Synth-U but in a multivariate setting with an extra attribute: **Multivariable**. The second variable of the time series is transferred from the first variable following four kinds of rules: X-axis Flip, Y-axis Flip, Shift Forward, Shift Backward. We present all the attribute information in Tab. 3.

We sample 1000 samples for 32 combinations of primary attributes (4 Trend Types × 2 Trend Directions × 4 Season Cycles) and get 32000 samples. We randomly split the samples into training set, validation set, and test set in a ratio of 6: 1: 1. Finally, we get 24000 training samples, 2400 validation samples, and 2400 test samples.

### A.1.1. PRIMARY ATTRIBUTES

The primary attributes are attributes that all data in a dataset have in common, such as trend and season in all time series. Such attributes are often represented in a structured format, but lack the ability to represent unique characteristics of the data.

**Trend Types**
There are 4 trend types: linear, quadratic, exponential, and logistic. As described above, $\mathbf{t}$ is used to obtain $\mathbf{x}$. For linear trend: $\mathbf{x}_{\text{trend}} = \mathbf{t}$, in this case $t_i \in [0, 1]$, $x_i \in [0, 1]$. For quadratic trend: $\mathbf{x}_{\text{trend}} = \mathbf{t}^2$, in this case $t_i \in [0, 1]$, $x_i \in [0, 1]$. For exponential trend: $\mathbf{x}_{\text{trend}} = \frac{2^{\mathbf{t}}}{1024}$, in this case $t_i \in [-10, 10]$, $x_i \in [0, 1]$. $\mathbf{x}_{\text{trend}}$ is needed to range from 0 to 1, so $t_i$ is in [-10,10]. For logistic trend: $\mathbf{x}_{\text{trend}} = \frac{1}{1+\exp(-\mathbf{t})}$, where $t_i \in [-10, 10]$, $x_i \in [0, 1]$. Similar to the exponential trend operation, we repeat a scaling process in logistic trend. To train the model more easily, $\mathbf{x}_{\text{trend}} = (\mathbf{x} - 0.5) \times 2$ is used to normalize $\mathbf{x}_{\text{trend}}$ to $[-1, 1]$.

**Trend Directions**
There are totally 2 directions: up and down. For instance, in the Cartesian coordinate system, a linear line from coordinates (0,0) to (1,1) represents an "up" trend, while another line from coordinates (0,0) to (1,-1) represents a "down" trend. To implement this idea, we used a simple method to generate these two types of data as the following formulas: up trend is $\mathbf{x}_{trend} = \mathbf{x}_{trend}$ and down trend is $\mathbf{x}_{trend} = -\mathbf{x}_{trend}$.

**Season Cycles**
The Season represents the number of cycles in a time series. We add [0,1,2,4] sinusoidal waves into the original synthetics time series.

$$\mathbf{x}_{season} = a\sin(2\pi t + \phi)$$

where $t \in [0, n_{cycle}], n_{cycle} \in [0, 2^0, 2^1, 2^2]$. The Random Variances of these two variables follow uniform distributions: $a \sim \mathcal{U}(0.4, 0.6), \ \phi \sim \mathcal{U}(0, 2\pi)$.

A.1.2. SECONDARY ATTRIBUTES

The secondary attributes are attributes that only some samples have. Such attributes are often hard to be represented in a structured format, containing sample-specific details.

**Local Shapelets**
We define three local shapelets: [single peak, sag, double peaks], which are used to simulate local details in real-world time series. A single peak consists of a linear rise and a linear fall, with a length of 9, 0 at both ends, the highest midpoint in the range of [1.0, 1.2]. Sag is a single peak with a downward concave shape that is symmetrical about the x-axis. Double peaks are two single peaks spliced together. We also divide the time series into three segments of equal length: the beginning, the middle, and the end, and add 0 or 1 shapelet to a random position in each segment. The probability of adding nothing is 0.7, and the probability of single peak, sag, and double peaks is 0.1.

**High Frequency Components**
High-frequency signals are common components in time series. We add high-frequency components to simulate real-world time series. The period of high-frequency components ranges from [0, 16, 32, 64] and is constructed in the same way as season where $a \sim \mathcal{U}(0.1, 0.3), \ \phi \sim \mathcal{U}(0, 2\pi)$.

**Noise**
The noises are added to simulate the real-world time series. Since noise is difficult to model, adding noise is more to increase the randomness and diversity of samples. Since noise is difficult to model, adding noise is more to increase the randomness and diversity of the sample. The noise will be sampled from a Gaussian distribution $\mathbf{x}_{noise} \sim \mathcal{N}(\mu, \sigma^2)$, where $\sigma \in [0.04, 0.06]$.

A.1.3. MULTIVARIABLE

The multi variable transfer rules include X-axis Flip, Y-axis Flip, Shift Forward, and Shift Backward. X-axis Flip and Y-axis Flip are flipping the time series of the first variable along the x-axis and y-axis to get the second variable. Shift Forward and Shift Backward are shifting the time series along the time dimension where the shift distance $d_{shift} \in [20, 40]$.

A.1.4. ATTRIBUTES TO TEXT DESCRIPTIONS

The text descriptions are generated from the attributes with prompt templates. Here are examples of text descriptions for each attribute in Synth-M dataset:

- Trend Type & Trend Direction: "The trend is up and linear."

- Season Cycles: "The season cycle is 4."

- High Frequency Component: "There is a high frequency component with a period of 64."

- Local Shapelet: "There is a sag at the beginning. The end part has double peaks."

- Multivariable: "Flip the variable 1 along x-axis can get variable 2."

A.1.5. RELEVANT AND IRRELEVANT TOKENS

As mentioned in Sec. 5.4.2, we manually annotated all tokens within the synthetic datasets, categorizing them as either relevant or irrelevant. Specifically, tokens associated with the attribute value (refer to Tab. 3) were labeled as relevant, while all other tokens were labeled as irrelevant. The results are presented as follows:

- Relevant tokens: "linear", "quad", "exp", "log", "up", "down", "peak", "peaks", "sag", "double", "single", "beginning", "end", "middle", "0", "1", "2", "3", "4", "5", "6", "7", "8", "9", "x", "y", "flip", "forward", "backward", "shift".

- Irrelevant tokens: "trend", "type", "direction", "cycle", "season", "the", "description", "time", "series", "variable", "area", "part", "has", "a", "and", "equal", ".", "is", "high", "frequency", "component", "with", "period", "along", "axis", "signal", "on", "at", "in", "of".

## A.2. Augmented real-world datasets

The Augmented real-world represents the datasets that are composed of real-world time series and manually generated text description, including the **ETTm1** and **Traffic**.

Specifically, the original datasets only contain time series data. We first employ `tsfresh` (Nils Braun, 2024) library to extract 6 time series features, serving as attributes for baseline input. Then, text annotations are generated from extracted features through prompt templates. Details are given in Sec. A.2.1. Thus, our processed datasets combine three types of data: time series data, text descriptions, and extracted attributes.

A.2.1. FEATURE EXTRACTION AND TEXT ANNOTATION

To annotate these real-world datasets with their corresponding text and attributes, we performed a series of processing steps, including feature extraction and text annotation. Specifically, we used the `tsfresh` (Nils Braun, 2024) library to extract four global features (Skewness, Kurtosis, Linear Trend Slope, FFT Frequency) and two local features (Local Linear Trend Slope and Number of Peaks) from the time series data. These features were then used to generate attributes and text descriptions. The extraction methods for each feature and the corresponding text generation process are detailed below.

| | Feature Name | Description |
|---|---|---|
| **Global Features** | Skewness | Measures the asymmetry of the time series value distribution. |
| | Kurtosis | Measures the sharpness of the time series value distribution. |
| | Linear Trend | Describes the overall trend direction and rate of change in the time series. |
| | FFT Frequency | Identifies the dominant periodicity in the frequency domain. |
| **Local Features** | Local Linear Trend | Captures the trend direction within each segment. |
| | Number of Peaks | Counts the number of local maxima within each segment. |

*Table 4.* Summary of global and local features extracted using `tsfresh` for augmented real-world datasets.

**Global Features Extraction and Text Annotation**

The global features describe the overall characteristics of the time series, which include:

**1. Skewness:** The skewness of the time series distribution is calculated through `tsfresh.skewness` which measures the asymmetry of the value distribution. Based on the skewness value:

- If skewness $< -0.5$, the text description is: "The distribution of the value in time series is shifted to the negative."

- If skewness $> 0.5$, the text description is: "The distribution of the value in time series is shifted to the positive."

- Otherwise, the text description is: "The distribution of the value in time series is symmetrical."

**2. Kurtosis:** The kurtosis of the time series distribution is calculated through `tsfresh.kurtosis`, which measures the sharpness of the distribution. Based on the kurtosis value:

- If kurtosis $< -0.5$, the text description is: "and has low kurtosis."

- If kurtosis > 0.5, the text description is: "and has high kurtosis."

- Otherwise, the text description is: "and has normal kurtosis."

**3. Linear Trend:** The overall trend of the time series is calculated through `tsfresh.linear_trend`, which describes the trend direction and intensity. For example:

- A strongly positive slope generates the description: "The time series is going up rapidly."

- A weakly positive slope generates the description: "The time series is going up slowly."

- A strongly negative slope generates the description: "The time series is going down rapidly."

- A weakly negative slope generates the description: "The time series is going down slowly."

- A near-zero slope generates the description: "The time series has no obvious trend direction."

**4. FFT Frequency:** The dominant frequency of the time series is identified through `tsfresh.fft_coefficient`. The frequency with the highest magnitude is used to generate the description: "The main season cycles is around $n$ pi," where $n$ is the index of the dominant frequency.

**Local Features Extraction and Text Annotation**

We divide the time series into three segments: beginning, middle, and end along the time dimension equally. The local feature extraction is performed on each segment independently, which includes:

**1. Local Linear Trend:** The local trend of the time series is calculated through `tsfresh.linear_trend`, which describes the trend direction and intensity. For example:

- "At the beginning, the time series slowly rises."

- "At the middle, the time series has no obvious trend direction."

- "At the end, the time series goes down rapidly."

**2. Number of Peaks:** The number of local maxima is counted using `tsfresh.number_peaks` with parameter `n=10`. This feature generates a description such as:

- "At the beginning, there are 3 peaks."

- "At the middle, there are 2 peaks."

- "At the end, there are 4 peaks."

After extracting all features, each time series is represented as a structured vector of indices, referred to as the attributes index. This vector includes:

- **Variable Index:** [0,$K$-1] where $K$ is the variable number.

- **Trend Attribute:** {0,1} which indicates {upward, downward}.

- **Seasonality Attribute:** [0,9] which indicates the dominant periodicity.

- **Skewness Attribute:** {0,1,2} which indicates {negative, positive, symmetrical}.

- **Kurtosis Attribute:** {0,1,2} which indicates {low, normal, high}.

The Text Caption for each time series is constructed by combining the descriptions of global and local features. For example:

- **Global description:** "The distribution of the value in time series is shifted to the negative and has high kurtosis. For the overall shape, the time series is going up rapidly. The main season cycle is around 2 pi."

- **Local description:** "At the beginning, there are 3 peaks, and the time series slowly rises. At the middle, there are 2 peaks, and the time series has no obvious trend direction. At the end, there are 4 peaks, and the time series is going up rapidly."

### A.2.2. ETTM1

**Original Dataset**

ETTm1 is obtained from the Electricity Transformer Dataset (ETDataset) (Zhou et al., 2021), a dataset specifically designed to address the challenges of long-sequence time series forecasting. It was collected through a real-world platform in collaboration with the research team and Beijing Guowang Fuda Science and Technology Development Company. The dataset covers a two-year period from July 2016 to July 2018.

ETTm1, as a subset of ETDataset, focuses on minute-level data collected from a single transformer station. Each data point in ETTm1 consists of 7 numerical features: High Useful Load (`HUFL`), High Useless Load (`HULL`), Middle Useful Load (`MUFL`), Middle Useless Load (`MULL`), Low Useful Load (`LUFL`), Low Useless Load (`LULL`), and the target variable Oil Temperature (`OT`).

**Processed Dataset**

The raw dataset, consisting of a single long sequence with a total length of 69,680, is first split along the temporal dimension into training, validation, and test sets in a ratio of 8:1:1, resulting in sample sizes of 55,744 and 6,968, and 6,968 across the original seven variables. Next, the seven-variate time series data are decomposed into univariate sequences. For each set, the long sequence is further divided into shorter sequences using a sliding window with a sequence length of 120 and a stride of 30. This process yields the final samples split: (train: 13,013, valid: 1,631, test: 1,631).

Till now, we obtain the time series dataset for each time series feature variable of size $(N, K, L)$, where $N$ is the number of samples, and $N_{\text{train}} = 13,013, N_{\text{valid}} = 1,631, N_{\text{test}} = 1,631$ for train, validation and test set. $K$ denotes the number of time series variables, and $K = 1$ in our processed dataset. $L$ represents the length of the time series, and $L = 120$ in this dataset.

Furthermore, we apply z-score normalization to each variable using the mean and standard deviation computed from the training set, which ensures that each variable has a mean of 0 and a standard deviation of 1, eliminating differences in scale across variables. After that, we perform the feature extraction and text annotation to obtain the attributes and the text descriptions, with processing details given in Appendix A.2.1.

The final text descriptions are of size $N$, where $N$ is the number of samples. Each sample contains 1 string, corresponding to the concatenated global and local descriptions of a time series sequence. The statistics of token numbers for the text descriptions are summarized in Tab. 5.

| Set | Average Tokens | Median Tokens | Max Tokens | Std. Dev. |
|---|---|---|---|---|
| Training | 107.28 | 107 | 131 | 7.49 |
| Validation | 107.00 | 107 | 127 | 7.63 |
| Test | 106.63 | 107 | 128 | 7.39 |

*Table 5.* Summary of token number statistics for ETTm1 dataset.

### A.2.3. TRAFFIC

**Original Dataset**

The Istanbul-Traffic dataset (Leo, 2024) provides minute-level time series data on Istanbul's traffic index. It includes three time series features: the Traffic index Overall(TI), Traffic index of Asian side(TI_An), and the Traffic index of European side(TI_Av). The dataset covers the time period from November 1, 2022, to June 16, 2024, with a sampling frequency of one minute and a weekly update frequency.

**Processed Dataset**

The raw dataset, consisting of a single long sequence with a total length of 817,769, is first taken a sample every ten minutes and split along the temporal dimension into training, validation, and test sets in a ratio of 8:1:1, resulting in sample sizes of 65421, 8178, and 8178 across the original 3 variables. Next, the three-variate time series data are decomposed into univariate sequences. For each set, the long sequence is further divided into shorter sequences using a sliding window with a sequence length of 144 and a stride of 24. This process yields the final sample split: (train: 8178, valid: 1023, test: 1023).

Till now, we obtain the time series dataset for each time series feature variable of size $(N, K, L)$, where $N$ is the number of samples, $N_{\text{train}} = 8178, N_{\text{valid}} = 1023, N_{\text{test}} = 1023$ for train, validation and test set. $K$ denotes the number of time series variables, and $K = 1$ in our processed dataset. $L$ represents the length of the time series, and $L = 144$ in this dataset.

Furthermore, we apply z-score normalization to each variable using the mean and standard deviation computed from the training set. After that, we perform the feature extraction and text annotation to obtain the attributes and the text descriptions, with processing details given in Appendix A.2.1.

The final text descriptions are of size $N$, where $N$ is the number of samples. Each sample contains 1 string, corresponding to the concatenated global and local descriptions of a time series sequence. The statistics of token numbers for the text descriptions are summarized in Tab.6.

| Set | Average Tokens | Median Tokens | Max Tokens | Std. Dev. |
|---|---|---|---|---|
| Training | 102.96 | 103 | 130 | 6.83 |
| Validation | 103.70 | 104 | 124 | 6.95 |
| Test | 101.90 | 101 | 126 | 6.64 |

*Table 6.* Summary of token number statistics for Traffic dataset.

### A.3. Real-world datasets

The real-world datasets represent the datasets that are composed of real-world time series and text description pairs, including the **BlindWays** and **Weather**.

Specifically, the original datasets contain paired time series and textual descriptions. We extract attributes from the original text using ChatGPT 3.5 (OpenAI, 2023) for baseline input, producing a processed dataset that combines three types of data: time series data, text descriptions, and extracted attributes.

### A.3.1. BLINDWAYS DATASET

**Original Dataset**

The raw data is sourced from the benchmark dataset BlindWays (Kim et al., 2024). It includes 3D motion data from 10 blind individuals and 1 visually impaired participant, capturing actions performed while navigating along 8 carefully designed urban routes using a white cane or guide dog, along with corresponding rich textual descriptions.

The motion data comprises multivariate data from 1,029 motion segments, recorded through 18 IMU sensors worn by each participant, detailing the positions, angles, and trajectories of 24 joints in the human body. Each joint is represented and recorded using three variables, resulting in a total of 72 variables.

Additionally, the dataset provides (2 * 1,029 = 2,058) detailed textual descriptions, which include high-level descriptions summarizing action intent and contextual background, and low-level descriptions that meticulously document action specifics, such as step count and cane usage, for each time series sample. We utilized the 1029 high-level descriptions as the textual descriptions paired with the time series in our work.

**Processed Dataset**

For the processed motion dataset, we conducted a random sampling on the raw dataset to partition the raw data into training, validation, and test sets. The 1,029 motion segments were split into a training set containing 823 samples, a validation set with 103 samples, and a test set also comprising 103 samples, resulting in a final distribution of (train: 823, validation: 103, test: 103) samples. Additionally, we extract attributes from the original text using ChatGPT 3.5 (OpenAI, 2023).

For the time series data, we maintain the same shape as the raw dataset. The shape of the time series data is represented as $(N, K, L)$, where $N$ denotes the number of samples. Specifically, for the training, validation, and test sets, we have

$N_{\text{train}} = 823$, $N_{\text{valid}} = 103$, and $N_{\text{test}} = 103$. The variable count $K = 72$, while the time series length $L = 600$. Furthermore, we apply z-score normalization to each variable using the mean and standard deviation computed from the training set.

For the text data, we utilize the high-level textual descriptions from the original BlindWays dataset. The text data has a shape of $N$, where $N$ indicates the number of samples, and we utilize the high-level textual descriptions in the original dataset. The statistic details of the number of tokens in the text data are given in Tab. 7.

| Set | Average Tokens | Median Tokens | Max Tokens | Std. Dev. |
|-----|----------------|---------------|------------|-----------|
| Training | 33.24 | 31 | 135.0 | 11.68 |
| Validation | 34.56 | 32 | 106.0 | 12.76 |
| Test | 32.51 | 32 | 71.0 | 10.23 |

*Table 7.* Summary of token number statistics for BlindWays dataset.

For the attributes, the shape is represented as $(N, A)$, where $N$ indicates the number of samples and $A = 2$ represents the number of attributes variables. The two attributes are derived by classifying and extracting features from the original dataset's text using ChatGPT 3.5 (OpenAI, 2023): the Guide Method and Hand. Guide Method attribute indicates the method employed by the blind individuals, encompassing two values: [cane, guide dog] (i.e. using a white cane or a guide dog). Hand attribute indicates the hand used by the blind individuals, with three possible values: [left, right, unknown] (i.e. left hand, right hand, or unknown).

| Attribute | Value |
|-----------|-------|
| Guide Method | [Cane, Guide Dog] |
| Hand | [Left, Right, Unknown] |

*Table 8.* Summary of attribute options for BlindWays dataset.

### A.3.2. WEATHER-CAPTIONED DATASET

**Original Dataset**

The raw data is derived from the weather station WS Beutenberg, located at the Max Planck Institute for Biogeochemistry in Jena, Germany. The dataset includes climate time series spanning 8 years from 2014 to 2022.

The time series data consist of 21 weather-related variables recorded at 10-minute intervals throughout the year, with timestamps accurate to the second. The 21 weather-related variables include: atmospheric pressure (p, mbar); temperature (T, degC); potential temperature (Tpot, K); dew point temperature (Tdew, degC); relative humidity (rh, %); maximum vapor pressure (VPmax, mbar); actual vapor pressure (VPact, mbar); vapor pressure deficit (VPdef, mbar); specific humidity (sh, g/kg); water vapor concentration (H2OC, mmol/mol); air density (rho, g/m³); wind speed (wv, m/s); maximum wind speed (max. wv, m/s); wind direction (wd, deg); rainfall (rain, mm); rain duration (raining, s); shortwave radiation (SWDR, W/m²); photosynthetically active radiation (PAR, μmol/m²/s); maximum photosynthetically active radiation (max. PAR, μmol/m²/s); logarithmic temperature (Tlog, degC); and carbon dioxide concentration (CO2, ppm).

Textual descriptions are generated using weather forecasts obtained from publicly available platforms. These descriptions are created using the GPT-4 model, which generates weather-related descriptions independently of the time series data, relying solely on the weather forecast information.

**Processed Dataset**

For the processed dataset, we initially aggregate samples over a 6-hour period and partition the aggregated samples in chronological order, using data from 2014 to 2020 for training, 2021 for validation, and 2022 for testing, resulting in 10,192 samples for training, 1,460 samples for validation, and 1,448 samples for testing. Each sample is a pair of time series data and text data. Additionally, we extract attributes from the original textual descriptions to serve as input for the baseline.

For time series data, the shape is represented as $(N, K, L)$, where $N$ is the number of samples, $K$ is the number of variables, and $L$ is the sequence length. Specifically, for the training, validation, and test sets, we have $N_{\text{train}} = 10,192$, $N_{\text{valid}} = 1,460$, and $N_{\text{test}} = 1,448$. The variable count $K = 21$, corresponding to the original weather variables. The sequence length $L$ is set to 36, representing 6 hours of data (10-minute intervals, 6 samples per hour, resulting in time series length of $6 \times 6 = 36$).

To standardize the data, z-score normalization is applied to each variable using the mean and standard deviation computed from the training set.

For text data, each time series sample is associated with three textual descriptions, allowing for variability in phrasing while maintaining the same semantic meaning. In the processed dataset, all descriptions are retained, and during training, one description is randomly sampled for each time series instance. The textual data has a shape of $N$, where $N$ is the number of samples and we choose 1 textual description from 3 available textual descriptions in each sample. The detailed statistics of the text length in tokens for the training, validation, and test sets are presented in Tab. 9:

| Set | Average Tokens | Median Tokens | Max Tokens | Std. Dev. |
|---|---|---|---|---|
| Training | 246.42 | 244 | 397 | 30.04 |
| Validation | 262.91 | 260 | 423 | 33.16 |
| Test | 258.61 | 256 | 388 | 31.49 |

*Table 9.* Summary of token number statistics for Weather dataset.

For attribute data, the shape is represented as $(N, A)$, where $N$ is the number of samples and $A = 7$ is the number of attributes. We use ChatGPT 3.5 (OpenAI, 2023) to classify and extract 7 attributes from the original textual descriptions, as detailed in Tab. 10:

| Attribute | Values |
|---|---|
| Season | spring, summer, fall, winter |
| Time of Day | early morning, morning, afternoon, evening |
| Weather Condition | sunny, cloudy, rain, foggy, snowy, unknown |
| Temperature Trend | increase, decrease, steady, unknown |
| Wind Direction | S, N, W, E, SW, SE, NW, NE, unknown |
| Atmospheric Condition | low, average, high, unknown |
| Humidity Level | low, average, high, unknown |

*Table 10.* Summary of attribute options for Weather dataset.

### A.4. Transformation between structured and unstructured condition

In this section, we summarize the transformation between structured attribute conditions and unstructured textual description conditions in the three aforementioned types of datasets.

For synthetic datasets (Synth-M, Synth-U), the entire datasets are manually generated. First, primary and secondary attributes are predefined for multi-semantic characteristics. Then, textual descriptions are generated by substituting attribute values into the prompt templates. Thus, we generate text from attributes, as described in Appendix A.1.4.

For augmented real-world datasets (ETTm1, Traffic), the original datasets only contain time series data. First, the attributes are extracted from the original time series using the `tsfresh`. Subsequently, the textual descriptions are chosen from textual templates determined by the value of the attributes. Thus, we generate text from attributes, as described in Appendix A.2.1.

For real-world datasets (BlindWays, Weather), the original datasets already contain time series-text pairs. The attributes are directly generated from the textual descriptions in the original datasets using ChatGPT 3.5 (OpenAI, 2023). Thus, we generated attributes from text for these datasets.

## B. Model Architecture

In this section, we will show the architecture of our method and some details about previously mentioned modules in Sec. 4.

**Model overview.** The VerbalTS model takes a pair of noisy time series $\mathbf{x}_t \in \mathbb{R}^{K \times L}$ and textual description $\mathbf{c} \in \mathbb{N}^M$ as inputs, and output the estimated noise $\boldsymbol{\epsilon}_t$. The noisy time series $\mathbf{x}_t \in \mathbb{R}^{K \times L}$ is patched into $R$ resolutions and encoded into embedding $\mathbf{P}_t \in \mathbb{R}^{K \times N \times D}$ via linear layers. TSA and SSA modules process $\mathbf{P}_t$ to compute temporal and spatial attention, followed by a patch decoder and a multi-resolution mixer, producing the estimated noise $\boldsymbol{\epsilon}_t \in \mathbb{R}^{K \times L}$, as mentioned in Sec. 4.1. Textual descriptions $\mathbf{c}$ are encoded into embeddings $\mathbf{E} \in \mathbb{R}^{M \times D}$ using a text encoder, followed by the multi-focal

text processor that generates a unified semantic representation $\bar{\mathbf{Z}} \in \mathbb{R}^{S \times K \times R \times D}$ for the complete generation process, as mentioned in Sec. 4.2. For guiding conditional generation, the text representation $\bar{\mathbf{Z}}$ is integrated with the time series representation $\mathbf{P}_t, t \in [1, T]$ via the custom element-wise multiplication and addition, as mentioned in Sec. 4.3.

## B.1. TSA and SSA attention mechanisms

As mentioned in Sec. 4.1, we introduce two attention mechanisms, Temporal Self-Attention (TSA) and Spatial Self-Attention (SSA), modeling the temporal and spatial dimensions of time series data, respectively. In TSA, we employ masking mechanisms to enforce intra-resolution attention for temporal modeling; While in SSA, the attention operation is applied across all variables, allowing every variable to interact with others.

Given a batch of noisy time series input $\mathbf{x}_t \in \mathbb{R}^{K \times L}$, multi-resolution patching and patch encoding are first performed, as mentioned in Sec. 4.1. Then, we obtain a multi-resolution embedding $\mathbf{P}_t \in \mathbb{R}^{K \times N \times D}$. $K$ denotes the number of variables, $N = \sum_{r=1}^{R} N_r$, where $N_r$ denotes the patch number for resolution $r$, and $D$ denotes the embedding dimension.

For simplicity and without causing confusion, we temporarily omit the notation $t$. The time series representation $\mathbf{P}$ is passed to the TSA, which is in the shape of $(K \times N \times D)$, Temporal Transformers compute the self-attention along the temporal dimension $N$, and in parallel along dimension $K$. Specifically, the dimension $K$ is regarded as the batch dimension, ensuring that each variable $k \in [1, K]$ with its time series embedding of length $N$ is independent of others. This allows attention computation to be performed independently for each variable.

To restrict temporal self-attention within each resolution $r \in [1, R]$, we leverage an attention mask $\mathbf{M} \in \mathbb{R}^{N \times N}$ that restricts attention to patches within the same resolution, where $R$ denotes the total number of resolutions. The mask is defined as:

$$\mathbf{M} = \begin{pmatrix} \mathbf{M}_1 & -\infty & \cdots & -\infty \\ -\infty & \mathbf{M}_2 & \cdots & -\infty \\ \vdots & \vdots & \ddots & \vdots \\ -\infty & -\infty & \cdots & \mathbf{M}_R \end{pmatrix},$$

where each sub-matrix $\mathbf{M}_r \in \mathbb{R}^{N_r \times N_r}$ with all elements set to $0$ allowing attention within resolution $r$:

$$\mathbf{M}_r = \begin{pmatrix} 0 & 0 & \cdots & 0 \\ 0 & 0 & \cdots & 0 \\ \vdots & \vdots & \ddots & \vdots \\ 0 & 0 & \cdots & 0 \end{pmatrix},$$

and the residual values are set to negative infinity to mask inter-resolution attention.

The temporal self-attention TSA output is computed as:

$$\mathbf{P}_{\text{TSA}} = \text{softmax}\left(\frac{\mathbf{Q}\mathbf{K}^\top}{\sqrt{D}} + \mathbf{M}\right)\mathbf{V},$$

Where $\mathbf{Q}, \mathbf{K}, \mathbf{V}$ are query, key, and value matrices calculated from $\mathbf{P}$. The output $\mathbf{P}_{\text{TSA}} \in \mathbb{R}^{K \times N \times D}$ is subsequently passed into the SSA module.

To facilitate the spatial attention computation along the variable dimension $K$ in SSA module, $\mathbf{P}_{\text{TSA}}$ is permuted into the shape of $(N \times K \times D)$, as the actual input of the Feature Transformer Layer to compute the self-attention along the spatial dimension $K$, and in parallel along dimension $N$. Similarly, the temporal dimension $N$ is regarded as the batch dimension, ensuring that each patch index $n \in [1, N]$ with its feature embedding of dimension $K$ is independent of others. This allows attention computation to be performed independently for each patch index, while each variable can have attention with every other variable effectively. The spatial self-attention output is computed as:

$$\mathbf{P}_{\text{SSA}} = \text{softmax}\left(\frac{\mathbf{Q}\mathbf{K}^\top}{\sqrt{D}}\right)\mathbf{V},$$

Where $\mathbf{Q}, \mathbf{K}, \mathbf{V}$ are query, key, and value matrices calculated from $\mathbf{P}_{\text{TSA}}$. The output $\mathbf{P}_{\text{SSA}} \in \mathbb{R}^{N \times K \times D}$ is subsequently passed into the FFN module.

The complete process for temporal and spatial modeling in the $j$-th layer of the noise estimator can be expressed as:

$$\mathbf{P}^{(j)} = \mathbf{P}^{(j-1)} + \text{FFN}(\text{SSA}(\text{TSA}(\mathbf{P}^{(j-1)} + \mathbf{e}))),$$

where $j \in [1, J]$, $\mathbf{P}^{(0)} = \mathbf{P}$, $\mathbf{e} \in \mathbb{R}^D$ is the embedding of the current denoising time step, and FFN represents the feed-forward network. The final output is $\mathbf{P}^{(J)} \in \mathbb{R}^{K \times N \times D}$.

## B.2. Patch Decoder and Multi-resolution Mixer

After applying the TSA, SSA and FFN in $J$ residual layers, we obtain the time series representation $\mathbf{P}^{(J)} \in \mathbb{R}^{K \times N \times D}$. To generate multi-resolution noise estimates, two output modules are required to process these embeddings into the final noise estimates, as mentioned in Sec. 4.1.

First, Patch Decoder divides the multi-resolution representation $\mathbf{P}^{(J)}$ into $R$ resolutions and decodes them back into the original time series space, generating estimated noises $\{\boldsymbol{\epsilon}^r\}_{r=1}^R$ for $R$ resolutions, where $\boldsymbol{\epsilon}^r \in \mathbb{R}^{K \times L}$.

Next, Multi-resolution Mixer combines the estimated noise at different resolutions output from the Patch Decoder, into the final multi-resolution noise estimate. Specifically, we concatenate the estimated noise from different resolutions $\{\boldsymbol{\epsilon}^r\}_{r=1}^R$, and feed them into a Multi-Layer Perceptron (MLP) to generate the final noise estimate:

$$\boldsymbol{\epsilon} = \text{MLP}\left([\boldsymbol{\epsilon}^1 \circ \boldsymbol{\epsilon}^2, \cdots \circ \boldsymbol{\epsilon}^R]\right),$$

where $\circ$ denotes the concatenation operation, $\boldsymbol{\epsilon} \in \mathbb{R}^{K \times L}$ denotes the final estimated noise.

## B.3. Custom Element-wise Multiplication And Addition

As mentioned in Sec. 4.3, we propose the custom element-wise multiplication and addition, denoted as $\odot$ and $\oplus$, for applying the condition. In this section, we will introduce the details of these two operations. Suppose we have two matrices $\mathbf{A} \in \mathbb{R}^{T \times K \times N \times D}$ and $\mathbf{B} \in \mathbb{R}^{S \times K \times R \times D}$, where $T$ is the number of the diffusion step, $K$ is the variable number, $N$ is the length of the patch sequence, $S$ is the stage number, and $R$ is the resolution number. Given each element $\mathbf{A}_{t,k,n} \in \mathbb{R}^{1 \times 1 \times 1 \times D}$ in $\mathbf{A}$ with index $t, k, n$, we can find the corresponding index $s, k, r$ of element $\mathbf{B}_{s,k,r} \in \mathbb{R}^{1 \times 1 \times 1 \times D}$ in $\mathbf{B}$ through the following rules:

$$s = \left\lfloor \frac{t \times S}{T} \right\rfloor,$$
$$k = k,$$
$$r = \arg\min_r \left[ (\Sigma_{i=1}^r N_r) \geq n \right],$$

where the first equation projects the given diffusion step $t$ to the corresponding diffusion stage $s$; the second equation projects the given variable $k$ to the same variable; and the third equation finds the corresponding resolution $r$ covering the given multi-resolution patch index $n$, as described in Sec. 4.1.

Then the custom element-wise multiplication and addition can be expressed as:

$$(\mathbf{A} \odot \mathbf{B})_{t,k,n} = \mathbf{A}_{t,k,n} \times \mathbf{B}_{s,k,r},$$
$$(\mathbf{A} \oplus \mathbf{B})_{t,k,n} = \mathbf{A}_{t,k,n} + \mathbf{B}_{s,k,r}.$$

With the custom element-wise multiplication and addition, we can apply the multi-semantic text conditions to the corresponding time series components.

# C. CTTP Model Details

As mentioned in Sec. 5.1, the Contrastive Text-Time Series Pretraining (CTTP) model is used to calculate the metrics which is conceptually similar to the CLIP model (Radford et al., 2021). The purpose of the CTTP model is to train the time series encoder $\psi_{\text{ts}}(\mathbf{x})$ and the text encoder $\psi_{\text{text}}(\mathbf{c})$ by learning the alignment between a batch of time series data $\mathbf{X} \in \mathbb{R}^{B \times K \times L}$ and its associated textual descriptions $\mathbf{C} \in \mathbb{N}^{B \times M}$, where $B$ is the batch size, paired samples $\mathbf{x}$ and $\mathbf{c}$ are individual samples

from $\mathbf{X}$ and $\mathbf{C}$, respectively. We use PatchTST (Nie et al., 2023) as the time series encoder and a pre-trained text model of Long-clip (Zhang et al., 2024b) as the text encoder. The output embeddings of the time series encoder and text encoder are $\mathbf{Z_x} \in \mathbb{R}^{B \times d}$ and $\mathbf{Z_c} \in \mathbb{R}^{B \times d}$, respectively. These embeddings are mapped to a shared embedding space, enabling the alignment of time series and text data. The model is trained to minimize the cross-entropy loss between $\mathbf{Z_x}$ and $\mathbf{Z_c}$. The pseudocode for the CTTP model is presented in Algorithm 1.

---

**Algorithm 1** Pseudocode for the CTTP Model

---

**Input:** A batch of time series and text pairs $(\mathbf{X} \in \mathbb{R}^{B \times K \times L}, \mathbf{C} \in \mathbb{N}^{B \times M})$
**Output:** Total cross-entropy loss $\mathcal{L}_{\text{cross}}$

1: **# Extract embeddings of time series and text (joint embedding space)**
2:    $\mathbf{Z_x} \leftarrow \text{TimeSeriesEncoder}(\mathbf{X})$                                      $\triangleright \mathbf{Z_x} \in \mathbb{R}^{B \times d}$, time series embeddings.
3:    $\mathbf{Z_c} \leftarrow \text{TextEncoder}(\mathbf{C})$                                               $\triangleright \mathbf{Z_c} \in \mathbb{R}^{B \times d}$, text embeddings.

4: **# Compute pairwise similarities**
5:    $\mathbf{S} \leftarrow \text{Sim}(\mathbf{Z_x}, \mathbf{Z_c})$                                     $\triangleright \mathbf{S} \in \mathbb{R}^{B \times B}$, similarity score matrix.

6: **# Compute cross-entropy losses**
7:    $\mathcal{L}_\mathbf{x} \leftarrow \text{CrossEntropy}(\mathbf{S}, \mathbf{I}, \text{axis} = 1)$                  $\triangleright$ Loss for time series alignment.
8:    $\mathcal{L}_\mathbf{c} \leftarrow \text{CrossEntropy}(\mathbf{S}, \mathbf{I}, \text{axis} = 0)$                       $\triangleright$ Loss for text alignment.

9: **# Total cross-entropy loss:**
10:    $\mathcal{L}_{\text{cross}} \leftarrow \frac{\mathcal{L}_\mathbf{x} + \mathcal{L}_\mathbf{c}}{2}$                                        $\triangleright$ Average loss.

11: **Return:** $\mathcal{L}_{\text{cross}}$

---

$d$ denotes the dimension of the joint embedding space, where both time series and text embeddings are projected. $\mathbf{I} \in \{0, 1\}^{B \times B}$ is the identity matrix.

The CTTP model effectively aligns time series with text using the cross-entropy loss, providing a shared embedding space. We further use this model to evaluate the quality of the generated time series.

Following contrastive learning practices (Radford et al., 2021), we evaluate the CTTP model using a retrieval-based protocol. For each time series in a random batch (B=32), we compute the top-1 accuracy of retrieving its paired text from among B candidates. The results in Tab. 11 demonstrate the CTTP model's effective semantic alignment ability, providing evidence of the metrics reliability.

| Dataset | Synth-M | BlindWays | Weather | Synth-U | ETTm1 | Traffic |
|---------|---------|-----------|---------|---------|-------|---------|
| Accuracy | 94.68% | 15.53% | 70.93% | 97.55% | 50.28% | 36.56 % |

*Table 11.* The evaluation results of CTTP models on different datasets. We report the accuracy of retrieving the most similar text description in 32 candidates given the time series.

## D. Evaluation Metrics

### D.1. FID

The Frechet Inception Distance (FID) (Heusel et al., 2017) is a widely used metric for evaluating the fidelity of generated data. FID is based on the Frechet Distance, which measures the similarity between two multivariate Gaussian distributions. By comparing the feature distributions of the real and generated data, it quantifies how closely the generated data resembles the real data.

Specifically, the FID is defined as:

$$\text{FID}(\hat{\mathbf{X}}, \mathbf{X}) = \|\mu_{\mathbf{z}} - \mu_{\hat{\mathbf{z}}}\|_2^2 + \text{Tr}(\mathbf{\Sigma}_{\mathbf{z}} + \mathbf{\Sigma}_{\hat{\mathbf{z}}} - 2(\mathbf{\Sigma}_{\mathbf{z}}\mathbf{\Sigma}_{\hat{\mathbf{z}}})^{\frac{1}{2}})$$

where $\mathbf{X} = \{\mathbf{x}_i\}_{i=1}^N$ and $\hat{\mathbf{X}} = \{\hat{\mathbf{x}}_i\}_{i=1}^N$ represent the sets of real and generated data samples with the size of $N$, respectively. The terms $\mu_{\mathbf{z}} \in \mathbb{R}^d$ and $\mathbf{\Sigma}_{\mathbf{z}} \in \mathbb{R}^{d \times d}$ are the mean and covariance of the real data embedding distribution $\{\mathbf{z}_i\}_{i=1}^N$, while $\mu_{\hat{\mathbf{z}}} \in \mathbb{R}^d$ and $\mathbf{\Sigma}_{\hat{\mathbf{z}}} \in \mathbb{R}^{d \times d}$ are the mean and covariance of the generated data embedding distribution $\{\hat{\mathbf{z}}_i\}_{i=1}^N$. $d$ is the dimensionality of the embedding. The embeddings $\mathbf{z}_i, \hat{\mathbf{z}}_i \in \mathbb{R}^d$ are constructed as:

$$\mathbf{z}_i = \psi_{\text{ts}}(\mathbf{x}_i), \quad \hat{\mathbf{z}}_i = \psi_{\text{ts}}(\hat{\mathbf{x}}_i)$$

where $\psi_{\text{ts}}(\cdot)$ is the time series encoder of the CTTP Model, which processes time series data $\mathbf{x}_i$ or $\hat{\mathbf{x}}_i$ to generate latent embeddings.

For the FID formula, the first term $\|\mu_{\mathbf{z}} - \mu_{\hat{\mathbf{z}}}\|_2^2$ measures the squared Euclidean distance between the mean vectors of the real and generated data distributions, reflecting the difference in their central tendencies. The second term, $\text{Tr}(\mathbf{\Sigma}_{\mathbf{z}} + \mathbf{\Sigma}_{\hat{\mathbf{z}}} - 2(\mathbf{\Sigma}_{\mathbf{z}}\mathbf{\Sigma}_{\hat{\mathbf{z}}})^{\frac{1}{2}})$, captures the discrepancy in the covariance structures of the two distributions, where Tr denotes the trace of a matrix. A smaller FID value indicates higher similarity between the real and generated data distribution, suggesting that the generated data has a higher fidelity.

### D.2. J-FTSD

The Joint Frechet Time Series Distance (J-FTSD) (Narasimhan et al., 2024) is a metric specifically designed to evaluate conditional time series generation models. J-FTSD measures the similarity between the joint distributions of real and generated data, incorporating both the time series and their associated conditions, which corresponds to the text in our setting. By calculating the Frechet Distance in the joint embedding space, it provides a comprehensive evaluation of the fidelity of the generated results, considering both the time series and text.

Specifically, the J-FTSD is defined as:

$$\text{J-FTSD}(\mathbf{D}, \hat{\mathbf{D}}) = \|\mu_{\mathbf{z}} - \mu_{\hat{\mathbf{z}}}\|_2^2 + \text{Tr}(\mathbf{\Sigma}_{\mathbf{z}} + \mathbf{\Sigma}_{\hat{\mathbf{z}}} - 2(\mathbf{\Sigma}_{\mathbf{z}}\mathbf{\Sigma}_{\hat{\mathbf{z}}})^{\frac{1}{2}})$$

where $\mathbf{D} = \{(\mathbf{x}_i, \mathbf{c}_i)\}_{i=1}^N$ and $\hat{\mathbf{D}} = \{(\hat{\mathbf{x}}_i, \mathbf{c}_i)\}_{i=1}^N$ represent the joint distributions of real and generated data with size of $N$, respectively. Each sample consists of a time series $\mathbf{x}_i$ or $\hat{\mathbf{x}}_i$ and its corresponding condition $\mathbf{c}_i$, which is text in the CTTP model. The terms $\mu_{\mathbf{z}} \in \mathbb{R}^{d_{\text{joint}}}$ and $\mathbf{\Sigma}_{\mathbf{z}} \in \mathbb{R}^{d_{\text{joint}} \times d_{\text{joint}}}$ are the mean and covariance of the joint embedding distribution of the real data $\{\mathbf{z}_i\}_{i=1}^N$, while $\mu_{\hat{\mathbf{z}}} \in \mathbb{R}^{d_{\text{joint}}}$ and $\mathbf{\Sigma}_{\hat{\mathbf{z}}} \in \mathbb{R}^{d_{\text{joint}} \times d_{\text{joint}}}$ are the mean and covariance of the joint embedding distribution of the generated data $\{\hat{\mathbf{z}}_i\}_{i=1}^N$. $d_{\text{joint}} = d_{\text{ts}} + d_{\text{text}}$ is the dimensionality of the joint embedding, where $d_{\text{ts}}$ and $d_{\text{text}}$ are the dimensionality of the time series embedding and the text embedding respectively. The embeddings $\mathbf{z}_i, \hat{\mathbf{z}}_i \in \mathbb{R}^{d_{\text{joint}}}$ are constructed as:

$$\mathbf{z}_i = \psi_{\text{ts}}(\mathbf{x}_i) \circ \psi_{\text{text}}(\mathbf{c}_i), \quad \hat{\mathbf{z}}_i = \psi_{\text{ts}}(\hat{\mathbf{x}}_i) \circ \psi_{\text{text}}(\mathbf{c}_i)$$

where $\psi_{\text{ts}}(\cdot)$ is the time series encoder of the CTTP Model, $\psi_{\text{ts}}(\cdot)$ is the text encoder in the CTTP Model, and $\circ$ denotes concatenation. A smaller J-FTSD value indicates higher similarity between the real and generated data, which comprehensively evaluates the fidelity of conditional time series generation by considering both the temporal and contextual similarity in the joint embedding space.

### D.3. CTTP Score

The CTTP score evaluates the sample-level alignment between the generated time series and the corresponding text description. It is calculated through the CTTP model, which aligns time series with text descriptions in a shared embedding space. Details of the CTTP model are provided in Appendix C.

Specifically, the CTTP model is used to encode the time series $\mathbf{x}$ to the embedding $\mathbf{z}_{\mathbf{x}} = \psi_{\text{ts}}(\mathbf{x})$, and encode the text $\mathbf{c}$ to the embedding $\mathbf{z}_{\mathbf{c}} = \psi_{\text{text}}(\mathbf{c})$, where $\psi_{\text{ts}}(\cdot)$ and $\psi_{\text{text}}(\cdot)$ are the time series encoder and the text encoder of the CTTP Model, respectively. The CTTP score is then defined as the dot product between $\mathbf{z}_{\mathbf{x}}$ and $\mathbf{z}_{\mathbf{c}}$, given by:

$$\text{CTTP}(\mathbf{x}, \mathbf{c}) = \mathbf{z}_{\mathbf{x}} \cdot \mathbf{z}_{\mathbf{c}}$$

where $\cdot$ denotes the dot product. A higher CTTP score indicates better semantic alignment between the time series data and its corresponding textual description, reflecting the CTTP model's ability to capture meaningful relationships between these two modalities.

# E. Implement Details

## E.1. Main Experiment

For all experiments, we set the number of diffusion steps as T = 50, embedding size for attributes and time series as 64. For training, we use Adam optimizer to train the model, the initial learning rate is set to be 1e-4 with MultiStepLR scheduler for all datasets, the batch size is set to be 512 for Synth-M, Synth-U, Weather, ETTm1 and Traffic, 16 for BlindWays. For the hyperparameters of the multi-focal modeling, $(R, S) = (3, 3)$ for all datasets. All our experiments were conducted three times running with different random seeds.

## E.2. Effect of Multi-focal Text Processing

To evaluate how VERBALTS processes nuanced textual details (Sec. 5.4.3), we analyze the averaged attention values of each token in the vocabulary. For each text description in the dataset, we first divide it into separate tokens and encode the tokens to embeddings independently. Then the token embeddings are used to calculate the attention values through Softmax$(\frac{\mathbf{Q}_A \mathbf{K}_E^\top}{\sqrt{D}})$ in Eq. (5). For each token in the vocabulary, we calculate the average of its attention values across all sentences in which it appears. It's worth noticing that, in Fig. 6, we position semantically relevant tokens in the first half of the x-axis and place irrelevant tokens in the second half. The attention value distribution across the vocabulary further demonstrates that VERBALTS learns to focus on semantically relevant tokens.

# F. Experimental Results

## F.1. More T-SNE Visualization Results

We provide more visualization results comparing the generation based on unstructured and structured data 5.4.1.

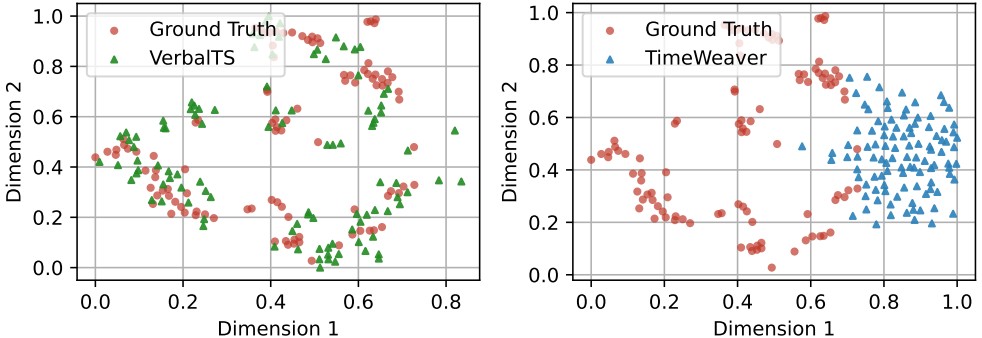

*Figure 9.* Comparison of generated data distribution between the VERBALTS (left) and TimeWeaver (right) on BlindWays dataset.

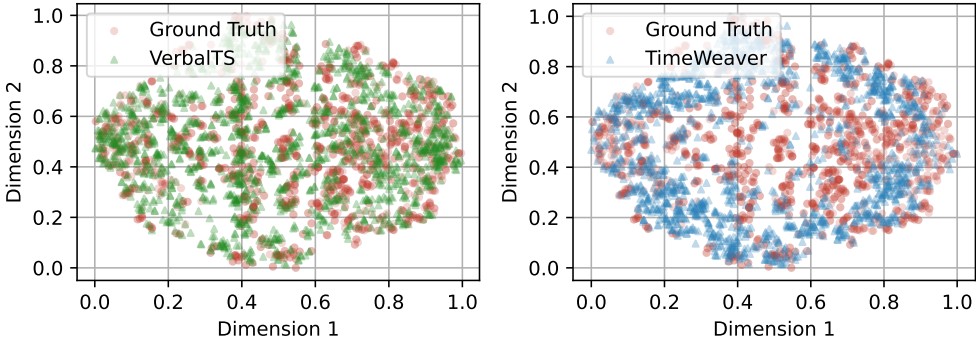

*Figure 10.* Comparison of generated data distribution between the VERBALTS (left) and TimeWeaver (right) on ETTm1 dataset.

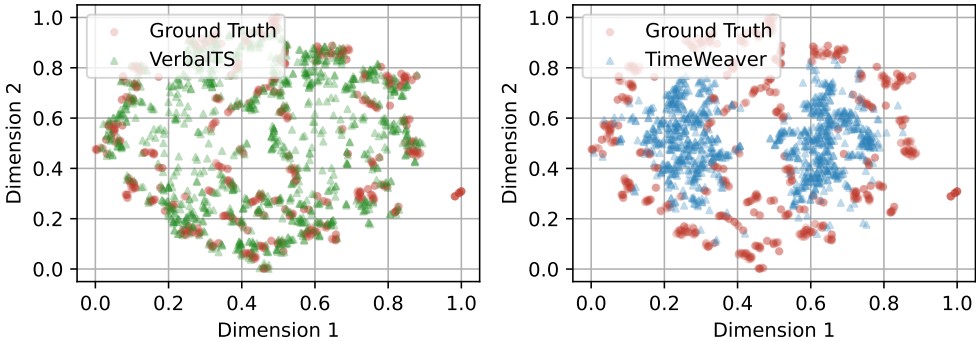

*Figure 11.* Comparison of generated data distribution between the VERBALTS (left) and TimeWeaver (right) on Traffic dataset.

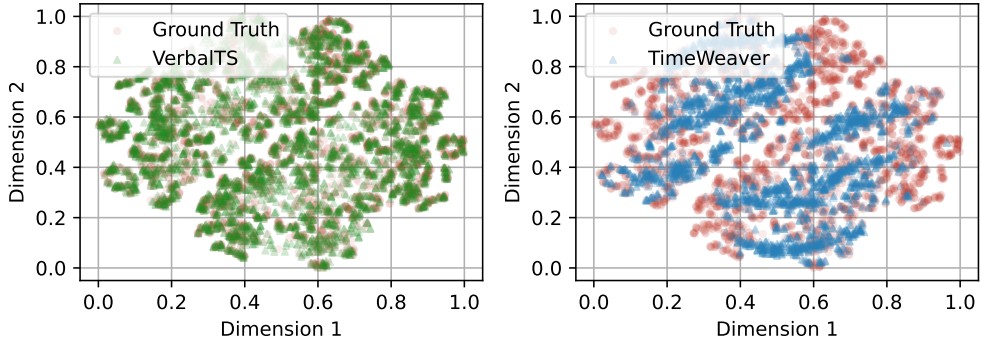

*Figure 12.* Comparison of generated data distribution between the VERBALTS (left) and TimeWeaver (right) on Synth-U dataset.

### F.2. Generation Case Study

In this section, we provide some visualization results (Sec. 5.2) to compare the generation ability of our method with the baselines. As shown in Fig. 13, VERBALTS better aligns with the semantic information, capturing nuanced details more accurately.

### F.3. Intervention Case Study

In this section, we present additional visualization results from the intervention experiment (Sec. 5.4.2), which compares the effects of masking semantically relevant tokens versus irrelevant ones. The results demonstrate that VERBALTS effectively focuses on semantically relevant information in the raw text to achieve fine-grained control.

### F.4. Editing Case Study

In this section, we provide more results of the editing experiment (Sec. 5.4.5) which compares the performance of VERBALTS with TEdit.

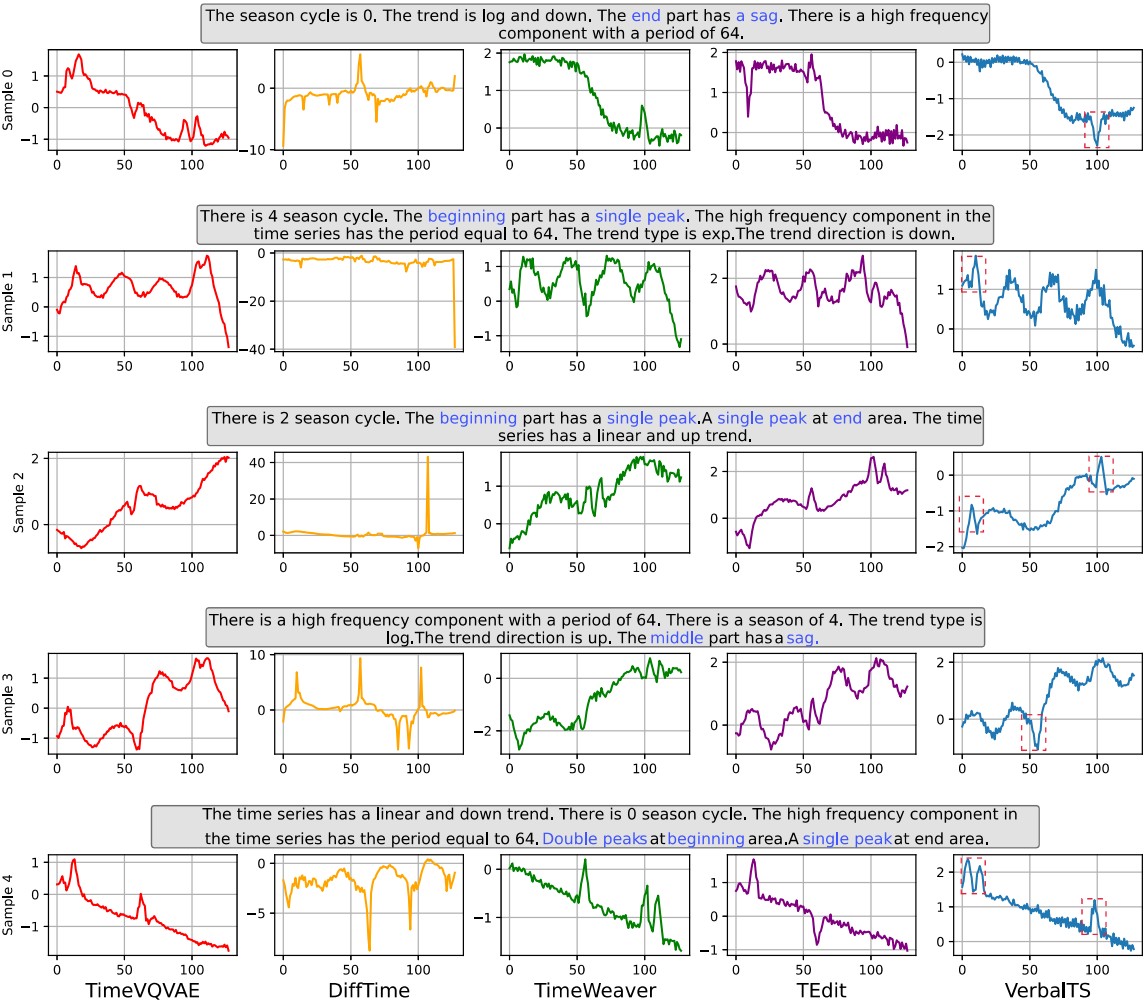

*Figure 13.* The qualitative comparison between the VERBALTS and baselines. From left to right column, there are TimeVQVAE, DiffTime, TimeWeaver, TEdit, and VerbalTS.

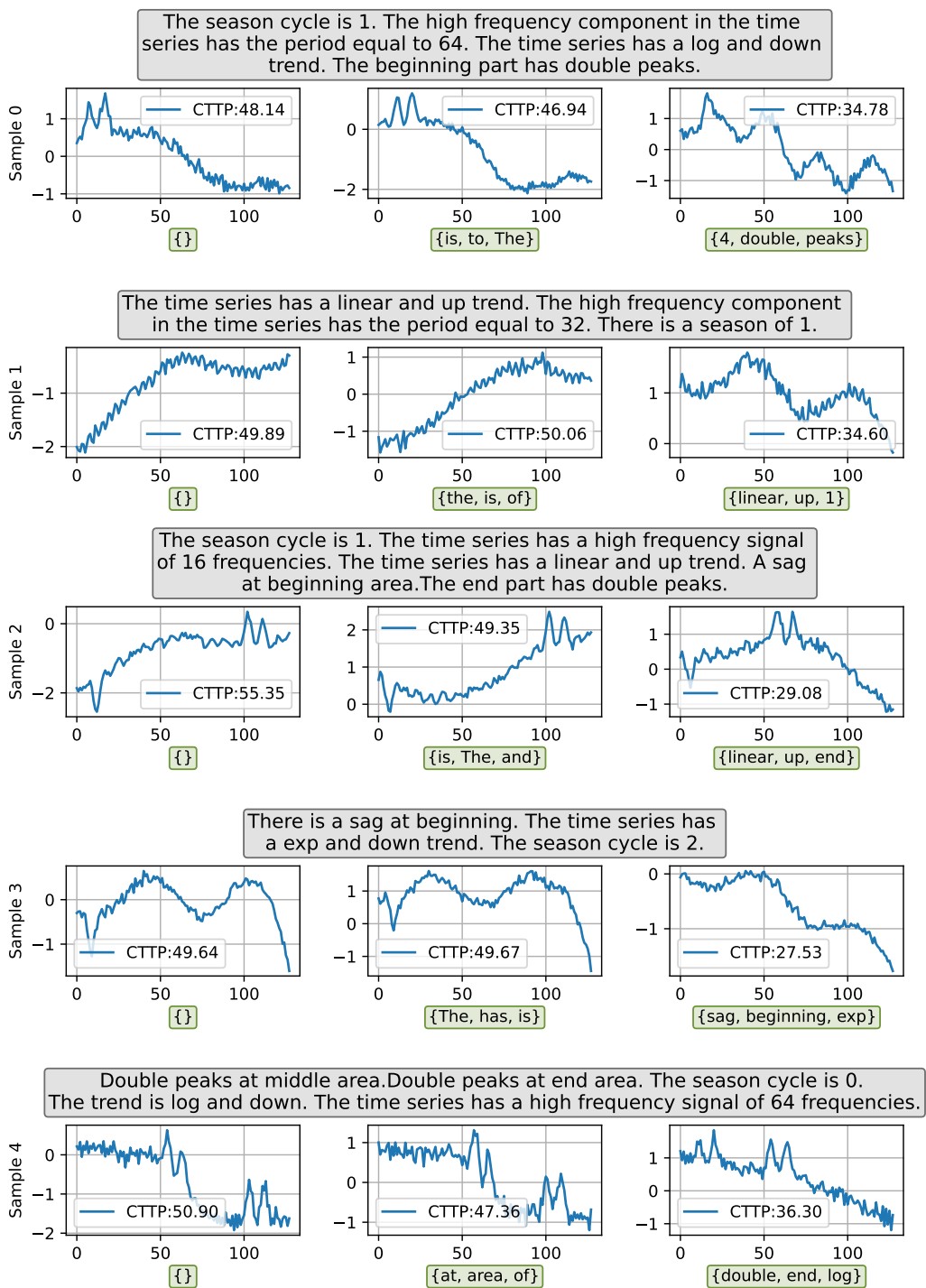

*Figure 14.* Comparison of generated data before masking (left column), after masking *irrelevant* tokens (middle column) and *relevant* tokens (right column). {Masked tokens} are shown below.

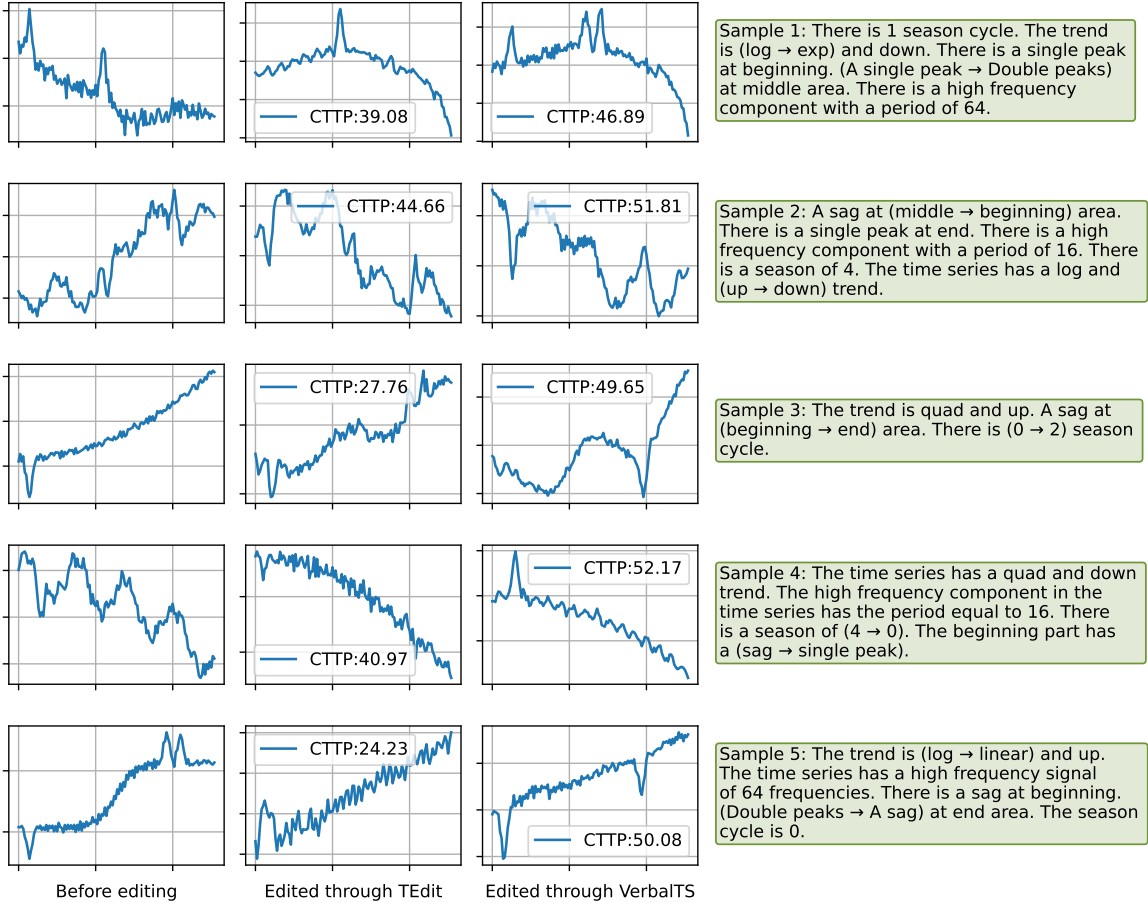

*Figure 15.* Illustration of editing task. Column 1: the raw time series before editing. Column 2: result edited by TEdit (Jing et al., 2024a); Column 3: result edited by our VERBALTS; Column 4: the condition prompts for editing, with (source → target) properties.

