# OpenReview forum: "VerbalTS: Generating Time Series from Texts"
_ICML.cc/2025/Conference — ICML 2025 poster_

### Official Review · Reviewer_Gv7f · 2025-03-08

**Overall Recommendation:** 3

**Summary:**

The paper introduces VERBALTS, a novel diffusion‐based framework that generates time series from unstructured textual descriptions rather than structured metadata. The authors argue that traditional time series synthesis is limited by the reliance on structured, expert‐annotated conditions and that text provides richer, sample‐specific control. VERBALTS leverages a multi‐focal text processor and multi‐view noise estimator to capture the hierarchical and multiresolution aspects of both text and time series. The approach is evaluated on two synthetic and (~four) real‐world datasets.

## Update after Rebuttal
I thank the authors again. The authors have mostly addressed my concerns. Unfortunately, the data pipeline is not included in the code, which is one of the key contributions of this work.

> While directly adopting vision models is feasible like PatchTST in forecasting, later works [1,2] indicate large improvement space. To our knowledge, our VerbalTS is the first study on text-to-ts generation. We believe further exploration beyond adopting vision models could lead to significant advancements on this novel task, and we hope our work may shed some light on it

This response mostly satisfies my question. However, I’m not sure if it is truly the first study on text-to-time-series generation—especially given the substantial body of research focused on generating trajectories for human avatars (the trajectories/time series of the joints). While I did not mention this in the first round, upon further reflection, I believe it’s important to acknowledge this line of work. The authors should consider including this in an updated version of the paper.

Tevet, Guy, et al. "Human motion diffusion model." ICLR 2023.

Shafir, Yonatan, et al. "Human motion diffusion as a generative prior." ICLR 2024.

I will maintain my score.

**Claims And Evidence:**

The entire paper is based on the claim that we cannot simply use image diffusion‐based methods such as Song et al. (2021) because "time series data exhibit multivariate characteristics and complex temporal dependencies (Torres et al., 2021; Wu et al., 2021), which fundamentally differ from the spatial structures typically encountered in image generation tasks." While I agree to a certain extent, later methods like PatchTST (Nie et al. ICLR 2023) have shown that one can essentially assume variate independence for time series‐based problems and utilize an off‐the‐shelf (vision) transformer; thus, simpler adaptations might be feasible. In addition, the authors use a text‐to‐image technique (Saharia et al. NIPS 2022) as a basis for their architecture, which partially contradicts their argument. If the authors persue further with this claim they need stronger evidence.

Otherwise, i think the remaining claims and research questions are properly backed and linked with evidence. The quantitative performance is measured in FID, J-FTSD, and CTTP, along with ablation studies that underscore the benefits of their framework.

**Essential References Not Discussed:**

The authors missed relevant time series modeling work that utilizes hierarchical/multiresolution techniques in their method—specifically, TS2Vec (Yue et al., AAAI 2021) and TimeMixer++ (Wang et al., ICLR 2025). Furthermore, they overlooked recent multimodal work based on LLMs, such as PromptCAST (Xue et al., TKDE 2023) and AutoTimes (Liu et al., NIPS 2024). Including these references would provide a more complete picture.

**Experimental Designs Or Analyses:**

I think they experiments are in general sound and are valid, however as I mentioned before the experiments are partially based on CTTP which needs in my opinion additional evidence (at least in the appendix).

**Methods And Evaluation Criteria:**

The proposed CTTP metric is an interesting approach for measuring the semantic alignment between generated time series and the input text.  Creating a new metric isn’t scientifically hard as long as the terms are clear that what’s being measured and you have external experts of domain aside from authors validating you.
While the definition of the metric is clear, the paper does not provide sufficient external validation or experiments to ensure that CTTP truly captures the intended properties. More insight into its calibration would strengthen the evaluation.

The authors use 2 entirely synthetic and 2 real world and 2 augmented real-world datasets, which support their findings.

**Other Comments Or Suggestions:**

This is not really an issue, but rather a suggestion:
There is an inconsistency in the color scheme across different plots, which is distracting and hampers the overall readability. A unified and clearly explained color scheme would greatly improve the visual presentation.
E.g. why is in Figure 1 the titles Blue as well as the time series, why look figure 3-6 look all different.

**Other Strengths And Weaknesses:**

## Strengths:
- S1. The integration of unstructured text for fine-grained control is novel for time series generation, offering a exciting approach to bypass the limitations of structured metadata.
- S2. The paper is well written, and the research questions are clearly stated and effectively linked to the experimental findings, which helps guide the reader through the contributions.
- S3. The augmented datasets and the detailed data pipeline presented in the work are valuable contributions for the time series community, as they address real-world data scarcity issues and offer new benchmarks for evaluation.

## Weaknesses:
- W1. Figure 2 is confusing, the different colors and color mixtures as well as the spatial ordering is very tough to get through, or at the very least, not adequately explained. This makes it challenging to quickly grasp the key insights when skimming through the paper.
- W2. Although the augmented datasets and data pipeline are significant contributions, a major drawback is that neither the model code nor the modified datasets are publicly shared. This limits reproducibility and hinders further research based on the work.

**Questions For Authors:**

1. Given the slower reverse diffusion process, what strategies do you have in mind for adapting VERBALTS to real-time or large-scale applications?
2. How sensitive is the performance of VERBALTS to the specific parameter choices, especially under noisy text conditions? Did you observe any situation where the performance greatly decreased?
3. Could you provide more evidence for your proposed CTTP Metric?
4. Would you mind sharing the code?

**Relation To Broader Scientific Literature:**

They build upon generating time series in order to address data scarcity for certain situations
(Narasimhan et al., 2024, Jing et al., 2024a). The architecture it self is based on an text2image method (Saharia et al. NIPS 2022).

**Theoretical Claims:**

The authors used a custom tailored Element-wise Multiplication And Addition operation which is further discussed in Appendix B.3

---

> ### Author Rebuttal · Authors · 2025-04-01
>
> Thanks for your recognition and valuable suggestions!
>
> > **Claims**: The paper's method contradicts the claim that can't simply use image diffusion‐based methods. The reviewer also think simpler adaptations might be feasible for text-to-ts (time series) generation.
>
> We apologize for any ambiguity. Our statement in Sec. 1—"generation framework based on diffusion models (Saharia et al., 2022) for text-to-ts generation"—refers to the general diffusion framework (Sec. 3.2), not a specific vision model. Our method introduces many innovations upon vanilla diffusion model, including multi-view noise estimator, multi-focal text processor and multi-modality semantic alignment (Sec. 4) to address challenges in text-to-ts generation. We'll clarify this in the revised paper.
>
> While directly adopting vision models is feasible like PatchTST in forecasting, later works [1,2] indicate large improvement space. To our knowledge, our VerbalTS is the first study on text-to-ts generation. We believe further exploration beyond adopting vision models could lead to significant advancements on this novel task, and we hope our work may shed some light on it.
>
> [1] iTransformer: Inverted Transformers Are Effective for Time Series Forecasting. ICLR 2024.
>
> [2] ElasTST: Towards Robust Varied-Horizon Forecasting with Elastic Time-Series Transformer. NeurIPS 2024.
>
> > **Evaluation**: More evidence of the CTTP's reliability are needed.
>
> Thanks for suggestion! We recognize the importance of metric reliability. Following contrastive learning practices [1], we evaluate CTTP model using a retrieval-based protocol. For each time series in a random batch (B=32), we compute top-1 accuracy of retrieving its paired text from B candidates. Results (https://files.catbox.moe/zf1a6s.png) show effective semantic alignment ablity of CTTP model. We’ll include this in the revised paper.
>
> [1] Learning Transferable Visual Models From Natural Language Supervision. ICML 2021.
>
> > **Reference**: Missing related works: TS2Vec, TimeMixer, PromptCAST and AutoTimes.
>
> Thanks for comments!
> + TS2Vec and TimeMixer++ only capture multi-scale patterns in time series, while our work tackles that in both text and time series modalities with cross-modal interactions.
> + Unlike LLM-based models (e.g., PromptCAST, AutoTimes) that use coarse-grained prompts to assist forecasting and QA, we leverages sample-specific text for fine-grained generation.
>
> We’ll include these works in the revised paper.
>
> > **W1**: Fig. 2 is confusing.
>
> We apologize for any confusion. To distinguish the three views and the information within them, we use different colors for variables, and color intensities for temporal resolutions. We'll follow other works like [1], simplify color scheme, clarify data flow for better readability.
>
> [1] Crossformer: Transformer Utilizing Cross-Dimension Dependency for Multivariate Time Series Forecasting. ICLR 2023.
>
> > **W2**: The datasets and code are not provided.
>
> Please check this link for codes: https://anonymous.4open.science/r/VerbalTS-E5BC.
>
> As is stated in Sec. 5 of our paper, we plan to release all the datasets and models upon the acceptance of the paper.
>
> > **Suggestion**: Unified and clearly explained color scheme should be considered, e.g., Fig. 1.
>
> Thanks!
> For Fig. 1, since we highlighted two different advantages of text in time series generation, we used two colors in it.
> We will refine Figs. 1 and 3-6 by unifying the legend style, marker sizes, and other visual elements to improve readability and consistency.
>
>
> > **Q1**: Any strategies for adapting VerbalTS to real-time or large-scale applications?
>
> It's a great point! The auto-regressive generation of reverse diffusion process is an efficiency challenge in practice. To address this, we have utilized DDIM [1] using deterministic non-Markov diffusion process with significant acceleration.
>
> Further improvements on diffusion efficiency include compressing the reverse steps via knowledge distillation [2] or performing diffusion in the latent space [3]. We plan to explore these to enhance our method for practice.
>
> [1] Denoising Diffusion Implicit Models. ICLR 2021.
>
> [2] Progressive Distillation for Fast Sampling of Diffusion Models. ICLR 2022.
>
> [3] High-Resolution Image Synthesis With Latent Diffusion Models. CVPR 2022.
>
> > **Q2**: How sensitive is VerbalTS to the specific parameters? Any situation where the performance greatly decreased?
>
> Thanks for comment. We added sensitivity study on Synth-M dataset to evaluate the impact of the multi-resolution number $R$ and diffusion stage number $S$, with results shown in https://files.catbox.moe/n6v94k.png.
> It suggests that, beyond certain threshold, the benefits of multi-resolution and multi-stage modeling become apparent. We plan to extend this analysis to more datasets and revise the paper accordingly.
>
> > **Q3**: More evidence for proposed CTTP metric.
>
> See response to Evaluation.
>
> > **Q4**: Would you mind sharing the code?
>
> Not at all! See response to W2.

---

### Official Review · Reviewer_aNaB · 2025-03-11

**Overall Recommendation:** 3

**Summary:**

This paper proposes VerbalTS, a novel framework for generating time series from unstructured textual descriptions. VerbalTS employs multi-focal alignment and generation framework that effectively models their complex relationships. Empirical evaluations demonstrate the benefits of VerbalTS in generation quality compared to existing baselines.

**Claims And Evidence:**

Generally fine.

Only concern is the evaluation size to support the claims in the "Extended Analysis"

**Essential References Not Discussed:**

N/A

**Experimental Designs Or Analyses:**

Checked all experiments; generally fine.

For the "Extend Analysis", the conclusion may not be solid enough, as the total evaluation datasets are not big, and these findings are based on the subset of the used datasets (e.g., Syn-M or Weather), which may not be strong enough to support the claim.

More concern please check question section,

**Methods And Evaluation Criteria:**

Generally fine.

Please see the concerns in the experimental design and question sections.

**Other Comments Or Suggestions:**

Minor suggestions:

1. The ablation study could better use w/o $A^s$,  w/o $A^t$, and  w/o $A^d$, to understand the each factors contribute to the improvements.

2. It is better to have some qualitative results or case studies that compare the generation difference between VerbalTS and baselines to understand the strength of the proposed method better.

**Other Strengths And Weaknesses:**

N/A

**Questions For Authors:**

1. The ablation study results show that even w/o $A^s$, $A^t$, and $A^d$, the generation results still outperform the baselines, e.g., on the Weather dataset. Could the author explain why this phenomenon, as it looks like the improvements are not exactly from the proposed model but the textual information.

2. Based on my first equation, I would want to understand whether the evaluations are fair to the baselines. Will the generation from text give more information than other conditions, particularly worried given that the textual information itself is generated from the original time series. Does that mean if you give specific enough textual descriptions of the data that comes from the data itself, it can somehow perfectly reconstruct the data? And should the paper compare with textual conditioned generation methods, if any?

3. In Table 1 results, ETTm1 and Traffic are known as multivariate time series datasets. Why do the evaluations on them show only univariate settings? It would be better to show both univariate and multivariate results on all four real-world datasets to show that the results are not cherry-picked.

4. Instead of using the generated text of ETT and etc., why not consider using more practical textual time-series pairs, such as doctor notes and medical time series, or several recent multi-modal time-series datasets (TimeMMD, ChatTime, etc.)

**Relation To Broader Scientific Literature:**

This work in time-series generation shows effectiveness in multiple scientific domains, as used in the evaluation.

**Theoretical Claims:**

N/A

---

> ### Author Rebuttal · Authors · 2025-04-01
>
> Thanks for the comments and suggestions!
>
> > **Exp**: Extended Analysis covers the subset of the used datasets.
>
> Thanks! The used datasets represent both synthetic and real-world datasets, showcasing diverse data coverage.
>
> Per your suggestion, we conducted additional experiments on other datasets, except those analysis requiring grounded generation function available only in synthetic datasets. The results (https://files.catbox.moe/dmmpmt.png) align with the findings in Sec. 5.4 of our paper. We'll revise the paper accordingly.
>
> > **Suggestion 1**: The ablation study could better use w/o $A^s$, w/o $A^t$ and w/o $A^d$.
>
> Per your request, we conducted an additional study by ablating each operation of the multi-focal text processing operators respectively, in https://files.catbox.moe/965rm7.png.
> The results indicate that all three operations in our method contribute to performance improvement, though the functions of different modules exhibit a certain degree of overlap in some dataset. Combining with the ablation study shown in Table 2 of paper, each operation plays a significant role in our method.
> We'll revise the paper accordingly.
>
> > **Suggestion 2**: Better show qualitative results or case studies.
>
> Thanks! We show some case studies of different methods in https://files.catbox.moe/1910op.png and will add this in revision. They demonstrate that our VerbalTS better aligns with the semantic information, capturing nuanced details more accurately.
>
> > **Q1**: The ablation study shows even w/o $A^s$, $A^t$ and $A^d$, the generation still outperform baselines. Why?
>
> Thanks for point.
> As discussed in Fig. 1, Secs. 1 & 4.2 and Finding 1 in Sec. 5.4, this improvement comes from more flexbly expressed textual conditions with nuanced time series details, highlighting the overlooked advantage of text-based conditional time series generation, as most studies focus on structured conditions like attributes or class labels, e.g., [1,2].
> Besides, our method further enhances performance (as in Tabs. 1-2, Finding 2, Fig. 4), showing potential improvements via better leveraging unstructured text in generation, indicating a promising direction for multi-modal time series research.
>
> > **Q2.1**: Will the generation from text give more information than other conditions?
>
> Yes, *unstructured* textual conditions may provide flexibly expressed and nuanced details that *structured* conditions cannot capture (discussed in Sec. 1).
> Yet, we believe the comparison is fair, as almost all the existing works rely on structured conditions. To our knowledge, this is the first study highlighting the impact of unstructured conditions on time series generation (Finding 1) and formally addressing this problem, as also recognized by Reviewer Gv7f. Our approach also well handles the noise in text (Finding 2) with enhanced multi-modal semantic alignment (Findings 3–5).
>
> > **Q2.2**: If given enough textual descriptions that comes from the data itself, can it perfectly reconstruct the data? Should the paper compare with textual conditioned generation methods, if any?
>
> It's a great point! However, as empirically observed, *perfectly* reconstructing a time series from texts remains challenging, as it's nearly impossible to describe time series perfectly. Yet, it's feasible to model the distributional alignment between text and time series modalities, which worth further exploration.
>
> To our knowledge, our work is the first on generating time series from *unstructured* texts, few works have attempted to this direction. Thus, we only compared with *structured* information based (e.g., metadata or attribute) conditional generation methods.
>
> > **Q3**: In Tab. 1, ETTm1 and Traffic are multivariate. Why show only univariate setting?
>
> This setting depends on dataset's textual conditioning property. For variate-specific textual descriptions (e.g., variate-specific text-augmented datasets ETTm1 and Traffic), we treat each variate independently to assess granular generation performance, following [1,2]. Contrastively, for texts describing multiple variates collectively (e.g., real-world datasets Weather and BlindWays), the multivariate time series is treated as a whole sample.
>
> > **Q4**: More practical textual time-series pairs like recently TimeMMD and ChatTime?
>
> Thanks! Beside the used 4 real-world datasets, more diverse datasets could further strengthen our experiments.
> Limited by rebuttal time, we conducted experiments on the Environment dataset in TimeMMD, its largest subset. The results (https://files.catbox.moe/uf3vs2.png) align with our original conclusion.
> Our focus is on fine-grained time series generation, those containing dataset-level descriptions like that in ChatTime are omitted.
> We'll add these to revise paper and continue incorporating more time series datasets.
>
> [1] Time Weaver: A Conditional Time Series Generation Model. ICML 2024.
>
> [2] Towards editing time series. NeurIPS 2024.

---

### Official Review · Reviewer_7E5h · 2025-03-15

**Overall Recommendation:** 3

**Summary:**

This paper studies time series generation, specifically generating time series from text. It proposes a method named verbalTS, which employs a multi-focus alignment and generation framework to effectively model the complex relationship between them.

**Claims And Evidence:**

NA

**Essential References Not Discussed:**

NA

**Experimental Designs Or Analyses:**

NA

**Methods And Evaluation Criteria:**

I think the main issue is that the authors omit several straightforward yet crucial baselines: directly using LLMs for time series generation, as well as employing visualizations for iterative revision. From my experience, the revision-based approach already achieves good generation performance. I suggest the authors incorporate multiple LLMs along with a simple iterative refinement mechanism as additional baselines.

**Other Comments Or Suggestions:**

NA

**Other Strengths And Weaknesses:**

No code.

I appreciate the authors' effort, and I believe time series generation is an interesting yet underexplored problem.

**Questions For Authors:**

NA

**Relation To Broader Scientific Literature:**

NA

**Theoretical Claims:**

NA

---

> ### Author Rebuttal · Authors · 2025-04-01
>
> Thank you for your valuable comments.
>
> > **Evaluation**: I think the main issue is that the authors omit several straightforward yet crucial baselines: directly using LLMs for time series generation, as well as employing visualizations for iterative revision. From my experience, the revision-based approach already achieves good generation performance. I suggest the authors incorporate multiple LLMs along with a simple iterative refinement mechanism as additional baselines.
>
> Thank you for your suggestions and great idea!
>
> **On LLM utilization for time series generation**:
> If our understanding is correct, your suggestion proposes leveraging large language models (LLMs) for time series generation with an iterative refinement mechanism, which is indeed an interesting idea. However, applying LLMs to time series generation presents several non-trivial challenges:
> + Understanding time series with LLMs remains an open problem [1]. As discussed in [1], fundamental reconsiderations are required for both the technical foundations of time series modeling and the associated evaluation and benchmarking methodologies.
> + Bridging the gap between text and time series modalities is challenging. LLMs are inherently designed for *discrete* text inputs and outputs, whereas time series data consists of *continuous* real-valued sequences. This discrepancy poses significant challenges in encoding and decoding time series using LLMs [2]. Our work introduces a novel approach to achieving fine-grained *semantic alignment* between text and time series, going beyond token-level reprogramming [2,3].
> + Limited paired text–time series data hinders progress in this direction. Few existing works explore general time series generation conditioned on unstructured textual information due to the scarcity of paired datasets. However, our work proposes a novel method to augment existing time series data with textual descriptions, a contribution also acknowledged in Strength 3 by Reviewer Gv7f.
>
> While leveraging LLMs for time series generation is not straightforward, it remains a promising avenue for exploration. Extending LLMs to model, understand, and generate time series beyond their traditional text-processing capabilities is an exciting research direction.
>
> **On the iterative refinement mechanism**:
> Your insight regarding the iterative refinement mechanism in the generation process is highly valuable! In our method, the multiple denoising steps in the reverse diffusion process naturally align with the concept of iterative refinement. Specifically, as described in Equation (3) of our paper, the model progressively refines the generated time series by iteratively denoising it at each step. The process begins with pure Gaussian noise and gradually converges to the fully generated time series. This autoregressive denoising mechanism closely resembles iterative refinement. A detailed explanation can be found in Section 3.2 of our paper.
>
> Overall, your proposed idea is highly intriguing. Exploring how multiple LLMs could iteratively generate time series is a promising research direction, and we plan to conduct more in-depth investigations and experiments in the future.
>
> [1] Are Language Models Actually Useful for Time Series Forecasting? NeurIPS 2024.
>
> [2] S2IP-LLM: Semantic Space Informed Prompt Learning with LLM for  Time Series Forecasting. ICML 2024.
>
> [3] Time-LLM: Time Series Forecasting by Reprogramming Large Language Models. ICLR 2024.
>
> > **W1**: no code.
>
> Thanks for your suggestion. You can find the details of our code at the following link: https://anonymous.4open.science/r/VerbalTS-E5BC
>
> As stated in Section 5 of our paper, we plan to release all datasets and models upon the paper's acceptance.

---

### Official Review · Reviewer_W3gc · 2025-03-15

**Overall Recommendation:** 3

**Summary:**

This paper presents a new task of generating time series data from unstructured text and introduces VERBALTS, a method that combines a multi-view time series noise estimator with a multi-focal text processor. Additionally, it establishes a new benchmark featuring multi-faceted time series datasets enriched with textual information.

**Claims And Evidence:**

yes

**Essential References Not Discussed:**

I think this paper have included all essential references in this research area.

**Experimental Designs Or Analyses:**

The experimental designs are reasonable and comprehensive.

**Methods And Evaluation Criteria:**

yes

**Other Comments Or Suggestions:**

please refer to weaknesses

**Other Strengths And Weaknesses:**

Strengths:
1. This paper introduces an interesting task of generating time series data from unstructured text and provides a corresponding benchmark.
2. Extensive experiments demonstrate that the proposed method achieves good performance.

Weaknesses:
The key aspect of this paper appears to be the alignment between text and time series data. However, the explanation of this process is unclear. It is not evident how different views and varying scales of time series data are aligned with the textual information. Additionally, the experimental analysis lacks an in-depth discussion on this critical aspect.

**Questions For Authors:**

please refer to weaknesses

**Relation To Broader Scientific Literature:**

The ideas presented could inspire further research into time series generation.

**Theoretical Claims:**

This paper does not contain any theoretical discussions and claims.

---

> ### Author Rebuttal · Authors · 2025-04-01
>
> Thank you for your valuable comments!
>
> > **W1**: The key aspect of this paper appears to be the alignment between text and time series data. However, the explanation of this process is unclear. It is not evident how different views and varying scales of time series data are aligned with the textual information. Additionally, the experimental analysis lacks an in-depth discussion on this critical aspect.
>
> In our article, we propose aligning time series and text descriptions from three views: the temporal view, the spatial view, and the diffusion view. Below we briefly explain these aspects with (i) motivation, (ii) technical approach and (iii) experimental evidence to make it clearer.
>
> (i) **Motivation**. This design of multi-view modeling and generation is upon the findings from existing studies and empirical observations:
> + From the temporal view, prior research [1] has shown that the influence of structured condition like attributes on time series generation spans multiple temporal scales as discussed in Section 4.1 in our paper. Similarly, in Section 5.4, we observe that the control of unstructured textual information over time series also operates across different time ranges and scales. For example, descriptions of trend direction influence the overall rise or fall of the time series, while descriptions of local peaks determine finer morphological details.
> + From the spatial view, we analyze the impact of text on different variables within the time series and find their influence varies across variables, as illustrated in Section 5.4. For instance, in motion generation, a description of "walking" predominantly influences the motion trajectory related to leg joint variables.
> + From the diffusion view, many studies like [2,3] have found that models at different diffusion steps capture information at varying granularities. For example, the early stages of denoising primarily shape the overall structure, while the later stages refine the details. We provide a comprehensive discussion of the motivation for our method in Sections 4.1 and 4.2.
>
> We hope the above contents help understand the motivation of our study.
>
> (ii) **Approach**. Based on the motivation mentioned above, we propose to achieve alignment between time series and textual information across these three views:
> + We first model the time series considering the three views. Specifically, we adopt a multi-resolution representation for the temporal view, employ spatial attention for the spatial view, and divide diffusion steps into multiple stages for the diffusion view. These are discussed in Section 4.1.
> + Next, we reprogram text descriptions into representations that correspond to these different views using a multi-focal text processor, as detailed in Section 4.2.
> + Finally, we align the multi-view text representations with the corresponding time series parts using a semantic adapter, as described in Section 4.3.
>
> We will keep refining the description of our paper to reflect the above summary.
>
> (iii) **Experimental Evidence**. In Section 5, we proposed three research questions first and carried out experimental analysis accordingly. Our method tackles the challenges of text and time series alignment and has significantly improve the text conditional time series generation, as shown in Section 5 and detailed as below:
> + In Section 5.3, the ablation study on the three views of text representation shows that incorporating multi-view alignment between time series and text descriptions enhances generation performance.
> + In Section 5.4.1, we demonstrated that multi-view alignment effectively alleviates negative influences of incorporated noise from unstructured text condition.
> + In Sections 5.4.2 and 5.4.3, we found that the multi-focal text processor mitigates the impact of noise in text, and the text processor assigns varying attention to tokens across different focuses.
> + In Section 5.4.4, we also showcased our method's effectiveness of semantic control on time series editing.
>
> We hope that the above summarization of experimental analysis has addressed your concerns about how our method tackles alignment between text and time series.
>
> In conclusion, we believe that our proposed method is well-motivated, conceptually sound, and clearly articulated in the paper and it's also been recognized by other reviewers (Strengths from Reviewers Gv7f and 7E5h). Furthermore, we have conducted comprehensive experiments to study this novel task of text-based conditional time series generation. We'll continue refining the paper to enhance clarity and readability, and we hope the explanation above provides a better understanding of our work. We look forward to further valuable discussions with you.
>
> [1] Towards editing time series. NeurIPS 2024.
>
> [2] How Control Information Influences Multilingual Text Image Generation and Editing? NeurIPS 2024.
>
> [3] MG-TSD: Multi-Granularity Time Series Diffusion Models with Guided Learning Process. ICLR 2024.

---

### Decision · Program_Chairs · 2025-05-01

**Decision:**

Accept (poster)

**Comment:**

This paper proposes a new task of generating time series data from unstructured text, and introduces VERBALTS, a method that combines a multi-view time series noise estimator with a multi-focus text processor.

The paper received four reviews, initially consisting of three weak accepts and one weak reject. The authors submitted a rebuttal, and after the discussion, all reviewers converged on a weak accept.

As the AC, I have read the paper, the reviews, and the rebuttal. I find that the paper presents an interesting and novel task and offers a reasonable solution. I agree with the emerging consensus among the reviewers and recommend acceptance.